# Hyaluronic Acid: Its Versatile Use in Ocular Drug Delivery with a Specific Focus on Hyaluronic Acid-Based Polyelectrolyte Complexes

**DOI:** 10.3390/pharmaceutics14071479

**Published:** 2022-07-15

**Authors:** Saoirse Casey-Power, Richie Ryan, Gautam Behl, Peter McLoughlin, Mark E. Byrne, Laurence Fitzhenry

**Affiliations:** 1Ocular Therapeutics Research Group, Pharmaceutical and Molecular Biotechnology Research Centre, Waterford Campus, South East Technological University, X91 K0EK Waterford, Ireland; richie.j.ryan@setu.ie (R.R.); gautam.behl@setu.ie (G.B.); peter.mcloughlin@setu.ie (P.M.); laurence.fitzhenry@setu.ie (L.F.); 2Biomimetic & Biohybrid Materials, Biomedical Devices & Drug Delivery Laboratories, Department of Biomedical Engineering, Henry M. Rowan College of Engineering, Rowan University, 201 Mullica Hill Road, Glassboro, NJ 08028, USA; byrnem@rowan.edu; 3Department of Chemical Engineering, Henry M. Rowan College of Engineering, Rowan University, 201 Mullica Hill Road, Glassboro, NJ 08028, USA

**Keywords:** hyaluronic acid, nanoformulations, ocular drug delivery, physicochemical properties, polyelectrolyte complexes, electrostatic interaction, artificial tears, comfort agent, mucoadhesion, viscosity enhancer

## Abstract

Extensive research is currently being conducted into novel ocular drug delivery systems (ODDS) that are capable of surpassing the limitations associated with conventional intraocular anterior and posterior segment treatments. Nanoformulations, including those synthesised from the natural, hydrophilic glycosaminoglycan, hyaluronic acid (HA), have gained significant traction due to their enhanced intraocular permeation, longer retention times, high physiological stability, inherent biocompatibility, and biodegradability. However, conventional nanoformulation preparation methods often require large volumes of organic solvent, chemical cross-linkers, and surfactants, which can pose significant toxicity risks. We present a comprehensive, critical review of the use of HA in the field of ophthalmology and ocular drug delivery, with a discussion of the physicochemical and biological properties of HA that render it a suitable excipient for drug delivery to both the anterior and posterior segments of the eye. The pivotal focus of this review is a discussion of the formation of HA-based nanoparticles via polyelectrolyte complexation, a mild method of preparation driven primarily by electrostatic interaction between opposing polyelectrolytes. To the best of our knowledge, despite the growing number of publications centred around the development of HA-based polyelectrolyte complexes (HA-PECs) for ocular drug delivery, no review articles have been published in this area. This review aims to bridge the identified gap in the literature by (1) reviewing recent advances in the area of HA-PECs for anterior and posterior ODD, (2) describing the mechanism and thermodynamics of polyelectrolyte complexation, and (3) critically evaluating the intrinsic and extrinsic formulation parameters that must be considered when designing HA-PECs for ocular application.

## 1. Introduction

Recent advances made in the development of novel therapeutic delivery systems for the treatment of ocular diseases affecting both the anterior and posterior segments can be seen in the review of Gote et al. [1]. Such delivery systems include, but are not limited to, thermosensitive in situ gelling systems for enhanced corneal adhesion [2,3], nanoformulations for enhanced therapeutic solubility, targetability and intraocular migration and retention [4,5,6,7], modified contact lenses for dual vision correction and therapeutic delivery [8,9,10,11,12], and bioerodible implants for the sustained, intravitreal delivery of anti-vascular endothelial growth factors [13,14]. Our understanding of the pathogenesis and origination of such diseases has also significantly improved, thus allowing for the development of targeted, stimuli-responsive delivery systems that exploit various routes of administration.

However, despite such successes, the global prevalence of moderate-to-severe visual impairment (MSVI) is continually increasing due to the combined effects of population growth and the increasing longevity of an ageing global population [15]. By 2050, the population of people over the age of 60 is set to reach approximately 2.1 billion [16,17]. As ageing is a primary risk factor for several moderate-to-severe visual abnormalities and ocular diseases, a concurrent exponential increase in MSVI cases is also expected. By 2050, it is estimated that over 474 million people will be suffering from some form of MSVI, a percentage increase of over 60% from 2020 [17]. In 2020, a total of 5.4% of the 206 million reported cases of MSVI within the global population of people over the age of 50 were attributed to ocular diseases such as age-related macular degeneration (AMD) (3%), glaucoma (2%), and diabetic retinopathy (1.4%) [18].

The global market for both therapeutics and drug delivery systems intended for the treatment of such diseases is set to expand exponentially in the coming years. It is predicted that the global market for therapeutics indicated in the treatment of AMD will reach USD 18.7 billion in 2028, an increase of USD 10.1 billion from the 2018 market value [19]. Due to increased demand for prostaglandin analogue-based formulations for the treatment of glaucoma in the ageing population, the global market for glaucoma treatment is predicted to reach USD 11 billion by 2027 [20]. Within the same timeframe, the global market for diabetic retinopathy is expected to reach USD 13 billion, predominantly due to the increasing prevalence of diabetes and the subsequent increase in hyperglycaemia-related retinal haemorrhaging in developed countries [21,22].

Posterior segment ocular diseases are primarily treated via intravitreal injections (IVI) or surgical intervention [23,24]. Such treatment options offer higher therapeutic retinal bioavailability in comparison to those administrated via topical or systemic administration [25]. A high concentration of the therapeutic is administered in the direct vicinity of the retina, thus bypassing the static and dynamic anatomical barriers within the anterior segment [26]. However, multiple administration of such injections over long periods can result in persistent, cumulative fluctuations in intraocular pressure (IOP), which can result in irreversible degeneration of the optic nerve [27]. Depending on the implantation site, surgical procedures involving delivery system implants pose the risk of moderate-to-severe adverse effects such as retinal haemorrhaging, acute inflammation, and ciliary body puncture [28].

Topical formulations such as eye drop formulations, ophthalmic gels, Ocuserts, and therapeutic-soaked contact lenses are non-invasive and can be self-administered in a non-clinical setting [29,30]. Treatments based on the topical administration of ocular therapeutics dominated the global ophthalmic market in 2020, equating to a total revenue share of over 62% [31]. However, selective inhibition of deeper intraocular therapeutic permeation through the static anatomical barriers, combined with precorneal clearance mechanisms such as the upregulation of tear film clearance via nasolacrimal drainage and reflex blinking, significantly reduces corneal retention time and therapeutic bioavailability at the target site [32]. Typically, only 3 to 5% of topically applied drugs penetrate the cornea and are available for further migration into the posterior intraocular tissues, whilst therapeutic bioavailability within the retinal tissues is highly dependent on the physiochemical attributes (molecular radius, lipophilicity) of the drug itself [33,34,35,36,37]. As such, to elicit a therapeutic response, repeated, more frequent administration may be required, thus resulting in the sporadic administration of the therapeutic in a manner that deviates from the recommended administration regime [38].

Polymeric ODDS such as nanoformulations, hydrogels, and biological stimuli-responsive systems have been developed to counteract the pitfalls of conventional ophthalmic formulations. Both synthetic and biological polymers have been used in the development of ODDs, with synthetic polymers exhibiting increased mechanical strength and stability in vivo, whereas the latter has the added advantages of biomimetic cell receptor targeting and enhanced biocompatibility resulting from fast in vivo enzymatic degradation. A thorough understanding of polysaccharidic and protein biopolymers such as chitosan, alginate, albumin, dextran, and gelatin can be seen in the reviews of Irimia et al. and Allyn et al. [39,40]. For the purpose of this review, hyaluronic acid (HA), a naturally occurring, high-molecular-weight (MW) glycosaminoglycan biopolymer, will be discussed in detail. HA has become a popular excipient in ocular drug delivery formulations as it possesses favourable characteristics such as pseudoplasticity, biocompatibility, water retention capacity, and biodegradability. Its tuneable viscosity and mucoadhesive properties are favourable for prolonging the precorneal residence time of various topical formulations. Its inherent hydrophilicity, attributed to the sequestration of water molecules within its network structure upon dispersion in aqueous media, allows for greater nanoformulation and therapeutic stability in vivo [41,42]. Targeted delivery in vivo is also achievable due to CD44 receptor-mediated endocytosis. Such prolonged and targeted therapeutic delivery at the target site is highly beneficial for reducing the frequency of administration and improving patient compliance, particularly for diseases affecting the posterior segment that are treated via invasive surgical procedures. However, HA undergoes rapid enzymatic degradation in vivo and may require physical or chemical modification to achieve therapeutically relevant stability in vivo [40].

## 2. Hyaluronic Acid: “Nature’s Sponge”

HA is a linear, non-sulphated glycosaminoglycan composed of repeating units of D-glucuronic acid and N-acetyl-D-glucosamine units [43,44,45]. Alternating β-1-3 glycosidic bonds link the individual monosaccharides, and β-1-4-glycosidic bonds link the disaccharide monomer units to one another [46]. HA has been branded ‘nature’s sponge’ due to its remarkable ability to absorb and retain water. When hydrated, HA has the capability of increasing its initial solid volume thousand-fold [47]. For this reason, it has become a primary component in several skincare and cosmetology products as a hydrating agent [48,49]. HA is a primary component of the extracellular matrices (ECM) surrounding connective tissues and exhibits inherent biocompatibility, biodegradability, and mucoadhesive capabilities [45,50,51,52]. It is found in abundance in synovial fluid, the umbilical cord and vitreous humour, at average concentrations of 2550 µg/cm^3^, 4100 µg/cm^3^, and 250 µg/cm^3^, respectively [46]. Such properties render it an ideal excipient for medical and pharmaceutical applications, such as artificial joint lubrication, hydrogel scaffolding for tissue engineering, and drug delivery systems.

### 2.1. Molecular Structure

As depicted in Figure 1, when both D-glucuronic acid and *N*-acetyl-D-glucosamine are in the β conformation, the functional moieties in the equatorial positions are sterically stabilised, whereas the hydrogen atoms on each monosaccharide unit occupy the more energetically and sterically unfavourable axial positions, forming the energetically stable conformation of HA [53,54,55]. Each disaccharide unit is inverted respective to the preceding disaccharide unit in the polysaccharide chain, allowing HA to adapt a coiled, yet semi-rigid, ribbon conformation in solution [56,57,58].

In aqueous solution, rigidity is further enhanced by the formation of intermolecular and intramolecular hydrogen bonds and solvent interactions [43,58]. The secondary, two-fold helical structure is stabilised by hydrogen bond formation between both the carboxylate and acetylamine moieties on neighbouring monosaccharide units and the equatorial hydroxyl groups and the oxygen atoms of the glycosidic bonds [59].

Mutual repulsion of the charged carboxylate anions and solvent–disaccharide interactions expand the rigid polysaccharide coils [58]. This expansion, coupled with the inherent rigidity of the chains, results in the formation of an expansive, hydrophilic domain and the entrapment of water molecules between the individual chains [60]. The MW of the individual HA chains is a critical factor in the reorganisation of such chains into a network structure. HA with a MW greater than 1600 kDa has exhibited enhanced polymer network formation in comparison to HA with a MW of less than 350 kDa, which has been attributed to increased chain entanglement and intermolecular hydrogen bond formation [61]. The hydrophobic regions of individual HA chains self-associate via supramolecular interactions such as hydrophobic interactions and intermolecular hydrogen bonds. Each of these supramolecular interactions contributes to the formation of the tertiary β sheet structure of HA [53]. This structure appears as an entangled, highly viscous network of individual HA chains aligned with one another in an anti-parallel configuration. Water molecules, electrolytes and nutrient molecules can freely diffuse throughout the domain. However, the diffusion of molecules with large hydrodynamic volumes, such as proteins and biopolymers, is hindered by the chain network. Tuneable diffusive properties can be beneficial for achieving sustained drug release from HA-based ODDs, particularly if the system is designed for the dual loading of therapeutic molecules with varying sizes.

### 2.2. Ionisation and Rheological Properties

HA is considered a weak polyelectrolyte. The degree of ionisation, or charge density, of the anionic carboxylate group is dependent on the pH of the reaction medium [62,63]. HA carboxylate anions possess a pKa of approximately 3 to 4 [64]. As such, at physiological pH, HA is negatively charged and can readily complex with cationic salts or cationic polyelectrolytes. The physicochemical state of HA, in relation to its ionisation, stability, and rheological behaviour, within specific pH ranges, is depicted in Table 1.

Aside from pH, the rheological properties of HA are also heavily dependent on intrinsic parameters such as MW and concentration. A direct correlation exists between both the MW and concentration of HA in solution and its intrinsic viscosity. However, the MW of HA is the primary deciding factor in the viscosity of the resulting solution. Endogenous HA, with a typical MW of approximately 1 × 10^6^ Da, exists as a highly cross-linked, entangled network, even at concentrations as low as 1 µg/cm^3^ [53]. Cowman et al. [69] demonstrated the inverse correlation between the MW and the concentration of HA on intrinsic viscosity. Upon evaluation of the intrinsic viscosity of HA solutions, 8600 µg/cm^3^ of 1 × 10^5^ Da HA was required to reach critical coil-overlap concentration. However, to reach critical coil-overlap within the sample prepared using 6 × 10^6^ Da HA, only 320 µg/cm^3^ was required. The specific volume occupied by HA increases with an increase in MW. Further expansion of the hydrophilic domain via electrostatic repulsion of an increased number of COO^−^ anions also contributes to an increase in both specific volume and intrinsic viscosity. Intermolecular hydrogen bond formation and hydrophobic self-association also increase with increasing MW, thus strengthening the rigidity of the entangled network and enhancing interchain entanglement.

At concentrations of between 1 and 4% (*w*/*v*), HA solutions form pseudo-gel structures with considerable intrinsic viscosity [46]. As the mass concentration in a given solution volume increases, intermolecular bonding and chain entanglement also increase. The individual HA chains cannot adapt configurations independently of one another and become fixed within the network. At a fixed HA MW, Bothner et al. [70] demonstrated an increase in HA zero shear viscosity with an increase in HA concentration from 10 to 20 mg/mL. At lower polymer concentrations, HA chains are weakly associated with one another and can behave independently of one another.

Fluctuations in extrinsic parameters, such as ionic strength and temperature, not only induce conformational changes but can also impact the charge density and rheological properties of HA in solution. The addition of cationic electrolytes, such as Na^+^, Li+, etc. can lead to a substantial decline in its intrinsic viscosity [46]. Cationic electrolytes suppress the electrostatic repulsion between the carboxylate anions on adjacent D-glucuronic acid units via the formation of extrinsic electrostatic pairs. Coupled with the weakening of the intermolecular hydrogen bonds, this results in the contraction of the dimensions of the established network domain, an increase in hydrophobic region self-association, and increased chain flowability [71]. Cations with a valence greater than two can behave as bridging molecules between two adjacent HA chains [43]. The formation of these intermolecular crosslinks results in the formation of a gel structure that attains the ability to retain large volumes of water.

The stability of the HA β-conformation is attributed to the steric hindrance imparted by the bulkier functional groups on the equatorial face of the disaccharide unit, which prevents free rotation about the β-1-4-glycosidic bonds. An increase in temperature allows for increased rotation, thus aiding chain mobility. The specific volume occupied decreases, resulting in a corresponding decrease in viscosity [71,72]. This increase in kinetic energy disrupts intramolecular and intermolecular hydrogen bond formation, thus hindering the stabilisation of the entangled network and promoting contraction via mutual aggregation.

The viscosity of HA-based ODDs can be fine-tuned depending on the preferred mode of administration. Viscous formulations are often preferred for topical administration to prolong the corneal residence times of the therapeutic. Conversely, formulations of low viscosity are preferable for posterior segment delivery due to the smaller injection force required for administration, thus preventing excess damage to the fragile posterior tissues [73].

### 2.3. Biological Properties of HA

#### 2.3.1. HA—A Fundamental Component of Vitreous Humour

The vitreous humour is composed of 98% water and 2% structural proteins [74]. A total of 75% of this structural protein network is composed of collagen type II [75]. The remaining 25% is composed of collagen type IX, HA, and other structural proteins such as fibrillin and opticin [76,77]. HA chains infiltrate the fibrillar structures composed of collagen type II and IX. A combination of water immobilisation and mutual electrostatic repulsion between HA chains results in the expansion of the collagen scaffold via an established swelling pressure. This ultimately increases the viscosity of the vitreous humour. The gelatinous nature of the vitreous humour imparts sufficient IOP on the retina to attain the overall spherical shape of the eye. HA is not uniformly distributed throughout the vitreous humour [78]. Higher concentrations of HA tend to accumulate within the posterior segment (1200 µg/cm^3^) whilst lower concentrations of approximately 250 µg/cm^3^ are present within the anterior segment [79].

#### 2.3.2. HA and Mucoadhesion

HA exhibits substantial mucoadhesive capabilities, readily forming hydrogen bonds with the MUC5AC glycoprotein oligomers within the mucoaqueous tear layer and the protective glycocalyx coating on the apical corneal epithelial cells [80]. Such binding forces significantly enhance the precorneal residence time of HA-based formulations.

#### 2.3.3. HA-CD44 Binding: An Essential Cellular Signalling Mechanism

HA can indirectly mediate several cell signal transduction pathways by binding to cell surface receptors such as Cluster determinant 44 (CD44), Toll-like receptors (TLR), and lymphatic vessel endothelial HA receptors (LYVE-1) [55]. CD44, a ubiquitously produced, transmembrane glycoprotein, is considered to be the primary HA receptor and its binding to HA directly influences a large number of physiological cellular processes. HA binds to specific amino acid residues on the extracellular HA-binding domain (HABD) of CD44. The C6 primary alcohol on the *N*-acetyl-D-glucosamine units binds to the tyrosine-109 residue of the HABD [81]. HA carboxylate anions also indirectly bind to the alanine 102 and 103 residues via the formation of water-bridged hydrogen bonds. Such interactions evoke conformational changes in the cytoplasmic domain of CD44, triggering several signalling cascades that are responsible for cellular proliferation, HA endocytosis, lymphocyte activation, inflammation, cytokine release, and cellular migration [45,82,83].

CD44-HA binding is of particular importance in corneal wound healing. The binding of HA to CD44 receptors within the corneal epithelium and endothelium layer stimulates cytoplasmic CD44 domain phosphorylation, which ultimately enhances fibroblast migration to the affected site [83,84]. HA can also facilitate corneal re-epithelisation and wound healing resulting from ocular damage via receptor-mediated activation of the pro-inflammatory response and angiogenesis. CD44 receptors are also expressed within the iris, conjunctival cells, and Muller glia within the retina under physiological conditions [60].

Increased proinflammatory cytokine expression and release are characteristic of the pathogeneses of several ocular diseases, such as AMD, glaucoma, and uveitis. Alternate splicing of CD44 messenger RNA (mRNA) in response to pro-inflammatory cytokine expression results in an increase in CD44 receptor expression [85]. Increased CD44 expression has been linked to the induction of choroidal neovascularisation in the retinal cells of a murine model post-laser-photocoagulation [86,87]. In primary open-angle glaucoma, a progressive decrease in HA concentration within the ECM of the trabecular network results in a corresponding increase in CD44 accumulation, a degradant of CD44 that is cytotoxic to retinal ganglion cells [88]. The overexpression of CD44 on the surface of ocular cells under pathological conditions presents a unique opportunity for the receptor-targeted delivery of HA-based ocular therapeutic systems.

#### 2.3.4. The Relationship between HA MW and Biological Activity

The biological activity of HA is highly dependent on MW. HA is endogenously produced within the inner plasma membrane via the enzymatic action of three isoenzymes, HA synthases 1, 2 and 3 [56,89]. The MW of the chains produced is controlled by the differential expression of the synthases in response to varying intracellular conditions. Native HA (NHA) has a MW of over 1 × 10^6^ Da and exhibits both anti-angiogenic and immunosuppressive properties [54,90,91]. NHA regulates the hydration, viscoelasticity, and structural integrity of the ECM via proteoglycan-mediated binding [53]. The exertion of swelling pressure due to the binding of NHA to water creates channels between adjacent tissues, creating an optimal environment for cellular migration. Low-MW HA (LMWHA) typically exhibits MWs of between 20 kDa and 500 kDa [45]. Polydispersity arises from NHA fragmentation via NHA synthases and degrading enzyme dysregulation under pathological conditions and via free-radical-mediated degradation [80]. LMWHA stimulates angiogenesis and cell mobility in response to cellular damage, enhancing wound healing and scar formation with the simultaneous activation of several pro-inflammatory mediators and growth factors [92,93].

The MW of HA also has an impact on the extent of CD44 receptor binding. NHA binds to CD44 via the formation of multivalent, cooperative interactions and has the capability of clustering CD44 receptors on the cellular surface, allowing CD44 to bind with secondary ligands [90,94]. Although the use of a high-MW HA will enhance the binding affinity of the therapeutic system to the target cells, it can also simultaneously induce adverse reactions that are stimulated by the transduction of the activated signal. This can prove detrimental to cells that are already under pathological stress. Through monovalent binding interactions, LMWHA and HA oligosaccharides (6–18 HA residues) bind to CD44 receptors with lower affinity in comparison to NHA [95].

### 2.4. Enzymatic Degradation of HA

An average-sized human has approximately 15 g of HA, 5 g of which is continuously turned over each day via enzymatic hydrolysis [95]. HA exhibits an approximate half-life of 10 weeks within the vitreous humour [55]. Hydrolysis of HA occurs via the enzymatic action of hyaluronidases (HYAL). HYAL-1 and HYAL-2 comprise the predominant hyaluronidases in somatic cells, cleaving the β-1-4-glycosidic bonds between the repeating units of D-glucuronic acid and *N*-acetyl-D-glucosamine via hydrolysis [44,83]. Rapid enzymatic degradation is unfavourable, particularly for HA-based ODDs that are designed for sustained therapeutic release within the posterior segment. As such, HA is often cross-linked via chemical modification to prolong its residence time and stability within the intraocular environment. The resulting fragments are internalised into lysosomes within the cytosol where HYAL-1 hydrolyses the fragments further, primarily forming tetrasaccharide residues [96]. Two exoglycosidases, β-glucuronidase and β-N-acetyl-D-hexosaminidase, cleave the tetrasaccharide residues further to yield monosaccharide units. Upon binding to CD44 receptors on the cell membrane surface, NHA and high-MW HA (HMWHA) are cleaved by HYAL-2, thus forming fragments of up to 20 kDa in size [92].

### 2.5. Potential Cytotoxicity of HA under Inflammatory Conditions

The relationship between HA MW and its inflammatory activity has been well documented in the literature, particularly in osteoarthritis models. The sequestration of multiple CD44 receptors via NHA and HMWHA-mediated divalent bonding suppresses both the expression of the pro-inflammatory cytokines and the formation of proteoglycans [97].

Conversely, LMWHA preferentially binds to TLR 2 and 4, transmembrane receptors that are primarily responsible for immune response initiation via pathogen-associated molecular pattern recognition [98]. The complexation of LMWHA and TLR triggers the nuclear translocation of the NF-κB protein, thus triggering the expression of several pro-inflammatory cytokines such as tumour necrosis factor-α (TNF-α), interleukin (IL)-1β, IL-6, and IL-8 [94]. As such, LMWHA fragments serve as potential biomarkers of cellular damage.

Under oxidative stress conditions, an inverse correlation exists between the concentration of HA and the chain length of the respective HA chains at the site of tissue injury [83]. The fragmentation of NHA via reactive oxygen (ROS) and nitrogen (RNS) species production, and imbalances in HA synthase and hyaluronidase activity, can result in excessive HA degradation via glycosidic bond cleavage, thus forming biologically active, pro-inflammatory HA fragments [99]. A vicious cycle between the mass production of ROS/RNS during tissue injury and pro-inflammatory cytokine activation via LMWHA-receptor binding is established, which may further amplify the inflammatory state and disease progression. It is also postulated that the reduction in thickness and viscosity of the HA protective barrier surrounding cells via ROS mediation contributes to both increased cell receptor accessibility and enhancement of the innate immune and inflammatory response [53].

Cytotoxicity can also arise due to the covalent modification of HA resulting from overexpression of tumour necrosis factor-stimulated gene-6 (TSG-6), an inflammation-associated secreted protein induced by TNF- α and IL-1, during inflammation [100]. TSG-6 catalyses the covalent bonding of the heavy chain (HC) domains of the chondroitin sulphate chains of IαI to HA, which results in the sequestration and aggregation of HA chains within a gelatinous hyaluronic acid–heavy chain (HA–HC) complex [71,101,102]. It is possible that HA–HC complexes can form within ocular tissues, due to the detection of both TSG-6 within human corneal epithelia and IαI within porcine trabecular meshwork cells, respectively [103,104]. The complexed aggregates are immobile and more viscous in comparison to NHA and, as such, exhibit enhanced resistance to biodegradation [105]. However, although such characteristics are favourable for the attenuation of joint degradation in arthritis, the intraocular sequestration of HA–HC aggregates may result in increased intraocular pressure and enhanced inflammation via the enzymatic and chemical fragmentation of localised HA within the complexes.

## 3. Hyaluronic Acid Biomaterials for Ocular Drug Delivery

Due to its tuneable physicochemical properties and inherent biocompatibility, the use of HA as an ophthalmic excipient has gained significant traction in recent years [51]. Many reviews have discussed the development of HA biomaterials, such as injectable hydrogels, nanofibrous membranes and films, and vitreous substitutes [59,60]. Similarly, recent studies have also demonstrated the efficacy in enhancing wound healing, protecting corneal epithelium cells, and attaining sufficient intraocular ocular lubrication and pressure during vitro-retinal and cataract surgery [3,42,79,106,107,108]. Therapeutic-laden nanomaterials and injectable hydrogels composed of HA have previously been injected into the posterior segment of the eye to both improve therapeutic accumulation at the target site and prolong its intra-retinal residence time [109,110,111,112]. Critical analysis of recent developments in HA-based ocular delivery systems is a primary focus of this review and will be discussed in greater detail in the following sections.

### 3.1. Artificial Tears for Dry Eye Disease

Artificial tear (AT) solutions are one of the primary treatment options to alleviate the discomfort and burning sensation caused by dry eye disease [113]. Dry eye is a multifactorial disease characterised by poor quality, unstable tear film secretion, primarily resulting from both the dysfunction of meibomian glands and loss of lubricating efficacy due to the progressive decrease in MUC5AC concentration within the mucoaqueous layer. Its pseudoplastic, rheological behaviour mimics that of the physiological tear film, thus allowing for increased flowability and distribution across the corneal surface during blinking upon application. The efficacy of AT solutions can be quantified by evaluating the stability of the tear film as a function of its break-up time.

You et al. [114] demonstrated the concentration-dependant therapeutic effect of HA-based AT solutions on tear film break-up time (TFBT) and conjunctival goblet cell stability in a dry eye-induced murine model. In comparison to the control groups, the 0.1%, 0.18%, and 0.3% (*w*/*v*) HA solutions significantly prolonged the TFBT, with the 0.3% (*w*/*v*) solution exhibiting superior efficacy by increasing TFBT by approximately 1.25 seconds. The 0.3% (*w*/*v*) solution also exhibited a protective effect against the deleterious effects of increased pro-inflammatory cytokine expression on conjunctival goblet cells.

Fallacara et al. [115] demonstrated the in vitro efficacy of a cross-linked HA and urea AT solution to stimulate corneal re-epithelisation upon induction of epithelial damage resulting from tear film dysregulation. After 72 h, the cells treated with the 0.02% and 0.4% (*w*/*v*) AT solutions exhibited similar viability of epithelial wound closure to that of the negative control. However, the corneal residence time of the AT formulations was not investigated as part of this study. Therefore, the results were inconclusive as to whether the crosslinking of carboxylate anions of HA with urea impacts its ability to bind to the mucin moieties on the corneal epithelium via hydrogen-bond-mediated adhesion. Graca et al. [116] evaluated the mucoadhesion of 0.15% and 0.30% (*w*/*v*) HA containing AT solutions using in vitro rheological studies. The 0.3% (*w*/*v*) HA formulation exhibited superior viscosity (382.2 mPa·s) when combined with hydrated porcine mucin in comparison to that of the 0.15% (*w*/*v*) formulation (53.2 mPa·s); this was attributed to the combined anionic charges of HA’s physiological pH and the negatively charged carbohydrate-bound ester sulphate residues and carboxyl groups on mucin oligosaccharides.

As a general note, none of the aforementioned studies included an investigation into the effect of MW on the viscosity of the HA formulations. As MW has a predominating effect on the intrinsic viscosity of HA in solution, this would be considered a critical parameter to investigate. Kojima et al. [117] demonstrated the increased efficacy of high-MW HA (HMWHA) in minimising the deleterious effects of dry eye in an environmental dry eye-stress murine model.

### 3.2. Eye Drop Formulations—Viscosity, Mucoadhesion, and Solubility Enhancement

Previous reviews have reported on the use of HA in therapeutic-laden eye drop formulations to both increase ocular residence time and enhance the therapeutic bioavailability and aqueous solubility of lipophilic therapeutics [52,118]. An increase in precorneal residence time ultimately reduces the rate at which the therapeutic is eliminated via nasolacrimal and conjunctival sac drainage. However, the viscosity of the formulation may be controlled to allow for sufficient mucoadhesion without adversely affecting vision clarity. Formulations should not exhibit an equivalent viscosity of greater than 30 mPa·s as the increased viscosity of the formulation may result in ocular discomfort and blurred vision [119].

Chen et al. [120] demonstrated the enhanced bioavailability and aqueous solubility of nimesulide, a non-steroidal anti-inflammatory drug, via conjugation to HMWHA (360 kDa) and LMWHA (36 kDa). Exhibiting equivalent tear osmolarity to that of normal human tears (296.7 ± 0.5 mOsm/L–LMWHA, 303.3 ± 4.2 mOsm/L–HMWHA), the nimesulide–HA conjugates significantly decreased interleukin 6 expression post-lipopolysaccharide-stimulated inflammation. Tredici et al. [121] also reported on the multi-factorial use of cross-linked HA as both a stabilising excipient and a therapeutic molecule to enhance in vivo corneal surface regeneration in a group of professional swimmers exposed to chlorinated water. Whilst therapeutic efficacy was proven, indicated by a reduction in tear osmolarity of 1.82 mOsm/L from baseline measurements, an investigation into the efficacy of the combination treatment in reducing the frequency of instillation was not conducted. Theoretically, HA should prolong the precorneal residence time of Coenzyme Q10 due to its mucoadhesive capabilities, enhancing its bioavailability whilst concurrently reducing the need for frequent administration. As shown in Figure 2, White et al. [122] demonstrated that HA exhibited superior water retention properties in comparison to the alternative polysaccharidic and acrylic comfort agents. HA also exhibited high values of zero-shear viscosity and flow viscosity with increasing MW, indicating its ability to reduce the rate of tear film drainage, attain tear film stability and volume, and maximise comfortability.

Jansook et al. [123] demonstrated the enhanced mucoadhesion of celecoxib and cyclodextrin ophthalmic solutions prepared with HA (MW 1000–1400 kDa). An increase in HA concentration from 0.1% to 0.5% (*w*/*v*) resulted in a significant increase in the concentration of celecoxib retained on the mucin-coated membrane. Salzillo et al. [124] also reported an increase in viscosity and mucoadhesion of the HA-based samples on simulated corneal–conjunctival epithelial mucin with an increase in HA concentration, with a more marked increase in such parameters with an increase in HA MW from 250 to 1120 kDa. Chen et al. [125] demonstrated the enhanced mucoadhesion of sodium hyaluronate (SH)–azithromycin eye drops in comparison to an azithromycin formulation containing hydroxypropyl methylcellulose (HPMC). The mean residence time of the SH formulation on rabbit eyes was approximately 1.56 times greater than that of the HPMC formulation, which they attributed to the combined viscoelasticity and bioadhesive nature of SH.

### 3.3. In Situ-Forming Hydrogel Systems

In situ-forming hydrogels, comprised of three-dimensional polymer networks cross-linked by physical or chemical bonds, exhibit remarkable swelling ability in aqueous media whilst attaining structural integrity [126]. Hydrogels exhibit remarkable swelling ability in aqueous media whilst retaining structural integrity [50]. Additionally, unlike the direct administration of therapeutics via IVI, hydrogel-based drug delivery systems can be designed to control and sustain the localised intraocular release of therapeutics, thus reducing the need for frequent administration. The use of bioerodible hydrogel systems also eliminates the need for subsequent surgical removal procedures, a quality that has become increasingly popular in the field of ophthalmology in recent years, particularly in the treatment of posterior segment pathologies [127]. Although HA exhibits physiochemical characteristics that are highly beneficial for hydrogel formation, it cannot form a physical gel by itself. [128]. Furthermore, its high elimination rate hinders its structural integrity in vivo [50,51]. Therefore, chemically modified or cross-linked HA derivatives are often used to formulate hydrogel systems intended for sustained therapeutic release.

#### 3.3.1. Injectable Formulations

Awwad et al. [129] prepared pNPAAM-based hydrogels, cross-linked with acrylated HA for the sustained release of infliximab and bevacizumab (BEVA) following IVI. Upon comparing in vitro enzymatic degradation, approximately 68% of the theoretical mass of the hydrogel containing 2.0 mg/mL of acrylated HA had degraded in 50 units per mL (U/mL) hyaluronidase over 4 days, whilst a weight loss of 41.5% was observed in the hydrogel prepared with 4.0 mg/mL of acrylated HA under the same conditions. This was attributed to the reduced enzyme ingression into the more densely cross-linked hydrogel. However, as a control pNIPAAM hydrogel was not analysed, it is difficult to evaluate the enhanced biodegradation capabilities of the hydrogels functionalised with HA.

Egbu et al. [130] reported the enzymatic and chemical crosslinking of in situ injectable hydrogels prepared with HA. Although the authors demonstrated a more sustained infliximab release over nine days from a pNIPAAM-HA hydrogel (24.9 ± 0.4%) in comparison to a 5% tyrosine-modified HA hydrogel (45.49 ± 0.3%), the reason why HA was added to the hydrogels prepared was unclear. HA may enhance the degradation and excretion of non-biodegradable pNIPAAM, thus improving the overall biocompatibility of the formulation. However, no in vitro degradation nor clearance studies were conducted to evaluate this hypothesis. Bora et al. [106] formulated a HA-based poly(D, L-lactide-co-glycide) (PLGA) microparticle-laden composite hydrogel system intended for the subconjunctival delivery of 5-fluorouracil. Optimal syringeability was obtained for the microparticle formulation that was dispersed in a 0.5% (*w*/*v*) solution of HA in comparison to those dispersed in solutions of lower concentration, attributed to the increased viscosity of the more concentrated HA solution. The in vitro release study highlighted the superior efficacy of the 4% methacrylated HA hydrogel in comparison to the 0.5% (*w*/*v*) solution in sustaining 5-fluorouracil release over 15 days, presumably due to the hindered diffusion of 5-fluorouracil through the hydrogel meshwork. Although capable of sustained release, the 4% methacrylated HA composite hydrogel was not characterised in terms of optical transparency, syringeability or rheological properties.

Yu et al. [131] conducted an in vivo release and biocompatibility study in which vinyl sulfone-functionalised HA hydrogels were administered to a rabbit eye via IVI. The hydrogels exhibited excellent cytocompatibility and exhibited similar retinal morphology and function to that of the control group over three months, whilst also attaining a BEVA concentration of approximately double the therapeutic concentration of BEVA after six months post-administration. The direct injection of BEVA into the pars plana attained a concentration of 2 × 10^−5^ ng/mL, a concentration that is 2.5 × 10^6^ times less than the therapeutically relevant dose.

#### 3.3.2. Vitreous Substitutes

Upon removal of the phase-separated vitreous during a vitrectomy, an artificial vitreous substitute is administered to both stabilise and tamponade the retina. A successful vitreous substitute must attain a density of 1.0071 g/cm^3^ and a refractive index of 1.3347, and exhibit porosity, biocompatibility, and stability equivalent to that of natural vitreous humour [132]. Silicone oil is a cornerstone vitreous substitute. However, due to the risk of emulsification post-administration, long-term intraocular cytotoxicity, and the need for a second removal surgery associated with silicone oil, alternative vitreous substitute components have been investigated in recent years [133]. HA-based hydrogels are of particular interest as they exhibit similar physicochemical properties to that of the natural vitreous humour.

A carboxymethyl chitosan-based in situ hydrogel, cross-linked with oxidised HA, for vitreous substitute application, was formulated by Wang et al. [134]. The hydrogels exhibited a comparable equilibrium water content, refractive index, and density to that of the physiological vitreous humour. As the HA concentration increased from 1% (*w*/*v*) to 4% (*w*/*v*), the aforementioned physicochemical properties also increased. The hydrogel preserved the original position of the retina 90 days after vitrectomy surgery whilst attaining a normal murine IOP of approximately 6 mmHg, with no adverse effects such as retinal detachment or opacification of the vitreous or haemorrhaging reported.

Thakur et al. [135] investigated the use of HA- and agar-based hydrogels for use as artificial vitreous humour in in vitro intravitreal drug delivery assays. The lower-viscosity HA–agar hydrogel, containing 0.7 mg/mL of HA, exhibited comparable sodium fluorescein neutral nanoparticle migration to that of excised bovine vitreous humour.

A glycidyl methacrylated HA hydrogel, cross-linked via the application of UV light at 365 nm in the presence of N-vinylpyrrolidone, was successfully formulated by Schulz et al. [136]. Although a slight decrease in viscoelasticity was observed upon passing the HA hydrogel through a 23G cannula, resulting from disruption of the hydrogel network under mechanical stress, the hydrogel attained similar viscoelastic behaviour to that of physiological vitreous humour.

### 3.4. HA-Based Topical Gels

HA-based hydrogels have also been developed as an alternative formulation to topical drug-laden ointments. Exhibiting the dual advantages of both an aqueous solution and an ointment, in situ hydrogels can prolong precorneal residence time due to their viscous nature whilst attaining optical transparency similar to that of natural vitreous humour [79].

Kim et al. [137] developed a 1,4-butanediol diglycidyl ether cross-linked HA membrane that exhibited a twofold increase in diameter after 5 days post-hydration. Transforming from a dry, rigid membrane to a soft, pliable hydrogel upon application, the hydrogel exhibited accelerated corneal re-epithelialisation and wound healing 72 h post-corneal-alkali-burn induction in a murine model. Interestingly, although the membrane was prepared using HA with an average MW of 700 kDa, the membrane released fragments ranging from 15 to 885 kDa. It was hypothesised that the wound healing induced by the membrane was attributed to the dual effect of both low- and high-MW HA fragments. Although it is known that both low- and high-MW HA exhibit regenerative properties, the preparation of a membrane with low-MW HA would have been beneficial to truly compare the in vivo efficacy against corneal injury.

Bao et al. [138] utilised oxidised HA with an average MW of 800 kDa and a glycol-chitosan and dexamethasone (DEX) conjugate to fabricate dual DEX and levofloxacin-laden hydrogel films for corneal application. One hundred percent of the theoretical levofloxacin concentration was released from the hydrogels of varying crosslinking density after 10 min. Conversely, following a burst release equating to 18% of the theoretical DEX concentration, a release of 720 µg of DEX was sustained over 24 h. Assuming a bioavailability of 10%, a single drop of DEX 0.1% (*w*/*v*) eye drop solution delivers 5 µg of DEX per application (30 µg a day with 6 applications). Therefore, via swelling and sustained diffusion mechanisms, the HA hydrogel was capable of delivering a therapeutic concentration of DEX that was 24 times higher than that of DEX 0.1% (*w*/*v*) eye drops in a single application.

Tomes et al. [139] synthesised an HA-based hydrogel to improve the aqueous solubility and corneal retention time of econazole, an antifungal therapeutic commonly prescribed for fungal keratitis. Solubilised with α-cyclodextrin, econazole release from the HA hydrogel was more sustained in comparison to the positive control econazole solution. Although the hydrogels exhibited a lower clearance rate and a mean residence time of 71 min on the ocular surface, viscosity and mucoadhesion studies would have been useful to demonstrate the superior retention capabilities in comparison to the econazole–cyclodextrin solution.

### 3.5. Contact Lenses: Comfort and Wettability via HA Modification

HA is an excellent comfort and wetting agent due to its water retention capabilities, a characteristic that is essential for the development of contact lenses designed for enhanced patient comfort. The addition of HA as a releasable wetting agent, as a conditioning agent in a soft contact lens care solution, or as an additive immobilised directly into the lenses via chemical crosslinking has demonstrated efficacy in increasing lens surface wettability, improving biocompatibility with the ocular surface, and preventing protein fouling [140].

Poly-2-hydroxyethyl methacrylate (pHEMA) lenses exhibit high water swellability and oxygen permeability due to co-polymerisation with hydrophilic monomers [141,142]. However, the addition of such monomers also increases surface protein deposition due to the electrostatic interaction-mediated bond formation between the electrochemically charged monomers and the cationic and anionic amino acid residues of lysozyme and albumin [143]. Van Beek et al. [144] demonstrated that immobilisation of HA within a pHEMA hydrogel via ethyl-3-[3-dimethylaminopropyl]carbodiimide hydrochloride crosslinking resulted in a significant decrease in both lysozyme and albumin surface absorption in comparison to the control. However, the optical transparency of the lenses decreased as the MW of the HA cross-linked into the pHEMA hydrogel increased due to the formation of large HA domains within the polymer matrix.

Silicone hydrogel (SIHy) contact lenses containing lipophilic silicone-based polymers such as poly(dimethyl siloxane) and tris (trimethylsiloxy) methacryloxy propyl silane (TRIS) are highly susceptible to lipid deposition and protein denaturation at the tear film–lens surface interface [145,146,147]. This, coupled with the inherent hydrophobicity and mechanical strength of SIHy lenses, contributes to decreased surface wettability, ocular irritation, and overall patient discomfort. Yamasaki et al. [148] compared the wettability of commercial SIHy lenses prepared with a low-MW, hydrophobically modified HA derivative (~10 kDa) to SIHy lenses prepared with 1000 kDa HA. Soaking the four commercial lenses in a multipurpose solution (MPS) containing the modified HA derivative (cleadew MPS) resulted in decreased contact angles in comparison to the lenses soaked in Biotrue^®^ (unmodified HA) and EOBO-41^®^ (polyoxyethylene-polyoxybutylene) after 13 cycles. The lower the contact angle, the higher the degree of wetting when the lenses come into contact with an aqueous solution [149]. Lower contact angles and lysozyme and human serum albumin deposition were also achieved by Korogiannaki et al. [140] by covalently attaching HA to the surface of a pHEMA lens co-polymerised with a TRIS monomer using thiol-ene “click” chemistry.

Utilising biomimetic molecular imprinting to create binding sites mimicking that of the CD44 receptor in daily disposable nelfilcon A lenses, Ali et al. [150] exhibited the dependence of HA release on both the percentage-by-mass of functional monomer and 2-(diethylamino) ethyl methacrylate (DEAEM) concentration. Additionally, utilising HA as a therapeutic, Maulvi et al. [151] designed UV photopolymerised pHEMA lenses containing an HA-laden ring implant. Although sustained release of HA in vivo was achieved over 15 days, pHEMA lenses are typically designed for daily, single-use application. As there was no significant difference between the degree of corneal staining in the positive control and HA-laden implant lenses group after one day post-application, the therapeutic efficacy of these lenses in promoting corneal wound healing may be limited. Additionally, a study to evaluate the effect of HA ring implantation on the critical optical properties of the pHEMA lenses was not conducted.

### 3.6. HA-Based Nanoformulations

The development of ophthalmic nanoformulations has been at the forefront of ophthalmology research in recent years as a means of overcoming the limitations associated with conventional ocular drug delivery systems [79,152,153]. The modification of such nanomaterials with HA presents unique advantages such as improved migration through the static anatomical ocular barriers, enhanced trans-membrane cellular permeation and enhanced bioadhesive capabilities through ligand-mediated CD44 receptor binding, and mucoadhesion [42,51,118]. Table 2 summarises the recent advances in the use of HA as an excipient in ophthalmic nanomaterial formulations and the impact of the addition of HA on the physicochemical attributes of the nanomaterials. As highlighted in Table 2, a variety of formulation methods are used in the synthesis of HA-based nanoformulations. However, preparation methods involving the use of desolvating agents, cross-linking moieties, large volumes of organic solvent, or high-energy formulation conditions to formulate nanoparticles with optimal physicochemical attributes that are suitable for mass scale-up are becoming increasingly unfavourable [154,155]. This is due to the potential cytotoxicity risks imparted by organic residues and the cost of the extensive purification steps to eliminate such residual material. The examples below highlight that HA-based nanoformulations prepared via self-assembly exhibit greater efficacy than alternative formulation methods in alleviating adverse conditions in both in vitro and in vivo ocular disease models, with the added advantage of mild formulation conditions.

#### 3.6.1. Enhanced Pharmacokinetics of HA-Based Nanoformulations

The addition of an HA coating can effectively enhance the intraocular migration of nanomaterials, particularly in the vitreous humour due to charge-mediated repulsion between the anionic carboxylate anions of HA and the negatively charged vitreal glycosaminoglycan moieties at physiological pH. HA-coated, mRNA-loaded lipoplexes of less than 100 nm exhibited enhanced mobility through bovine vitreous humour in an ex vivo model in comparison to the positively charged, uncoated lipoplexes [168]. The coating of the lipoplexes with HA with MWs of 22 and 2700 kDa did not trigger mRNA release in biological media, thus preserving the initial mRNA complexation efficiency of approximately 100%.

Conversely, the adsorption of 120 kDa HA onto the surface of positively charged, human serum-albumin nanoparticles resulted in a decrease in fluorescently labelled connexin43 mimetic peptide (Cx43-MP) encapsulation efficiency from 98.4 ± 0.1% to 85.4 ± 3.7%, with a concurrent reduction in zeta potential from 11.4 ± 0.2 mV to −18.2 ± 0.7 mV. The chemical conjugation of HA to nanoparticles, in which CX43-MP was loaded via an incorporation method, also resulted in a reduction in Cx43-MP encapsulation efficiency from 79.0 ± 1.9% to 71.1 ± 0.8%. However, the chemically modified nanoparticles also exhibited a highly negative surface charge (−44.0 ± 0.4 mV), which ultimately enhanced penetration through both the outer nuclear and RPE layer of an ex vivo bovine retinal model after 4 h of incubation. Conversely, the intraretinal diffusion of the uncoated nanoparticles was limited to the ganglion cell layer after 2 h [159]. In comparison to uncoated gold nanoparticles, HA-coated gold nanoparticles exhibited enhanced diffusion from the vitreal injection site within 4 h post-administration in an ex vivo porcine model [165]. The coated nanoparticles also maintained their inherent red-brown colour within the vitreous humour for up to 24 h, whereas the uncoated nanoparticles began to aggregate, indicated by a red-brown to purple colour change over the same period [158]. Ex vivo corneal permeability in a murine model increased significantly upon integrating HA within and around liposomal vesicles [169]. Hydrating a thin layer of soybean phosphatidylcholine with 0.7% (*w*/*v*) HA resulted in the spontaneous formation of gel-integrated liposomes. As the concentration of HA was fixed at 0.7% (*w*/*v*), it was unclear how the addition of HA affected fluconazole (FLZ) encapsulation efficiency. However, FLZ encapsulation efficiency increased from 25.8 ± 4.9% to 56.8 ± 2.5%, with an increase in FLZ loading from 0.3 to 1.2% (*w*/*v*). Corneal permeation was enhanced following HA modification, with the HA-modified liposomes allowing for the permeation of over 350 µg/cm^3^ of FLZ, an amount that was 1.9 and 2.6 times higher than that of the unmodified liposomes (~200 µg/cm^3^) and FLZ suspension (~140 µg/cm^3^), respectively. Stabilisation of the liposome core via HA gel integration also contributed to the sustained release of FLZ in vivo. The creation of a highly viscous, mucoadhesive diffusion network within the liposomes allowed for the sustained delivery of FLZ at a concentration above its minimum inhibitory concentration of 8 µg/mL for over 24 h post-instillation in an in vivo murine model. Conversely, the FLZ suspension reached a Cmax of 60 µg/mL 2 h post-instillation, followed by a rapid reduction in FLZ permeation after 6 h. Methyl ether poly(ethylene) glycol and 3-amino-1-propanol block copolymer micelles coated with HA exhibited a 1.5-fold increase in genistein corneal penetration in comparison to a genistein eye drop formulation in an ex vivo murine model after 10 h post-administration; this was attributed to the increase in precorneal retention time via hydrogen-bond-mediated mucoadhesion between HA and mucin chains [170].

#### 3.6.2. Cytotoxicity and Safety of HA-Based Nanoformulations

Although HA itself plays a significant role in corneal wound healing via enhanced corneal epithelium proliferation, the cytotoxic and safety evaluation of HA-based nanoformulations for ocular drug delivery remains an imperative step in the development process [107,115,171]. Kalam et al. [157] developed HA-coated CS nanoparticles that exhibited refractive indices (1.33 ± 0.08) and viscosity values (31.25 ± 2.05) that were suitable for topical ocular drug delivery without evoking adverse reactions such as blurred vision, nanoparticulate sensitivity, and discomfort. Huang et al. [162] developed an eye drop formulation laden with gelatin nanoparticles containing 20 µg/mL of EGCG and 62.5 µg/mL of HA. The nanoparticles exhibited an osmolality value (291 ± 11.1 mOsm/kg) that was within the acceptable range for normal human tear osmolality. An HA concentration-dependent increase in human corneal epithelial (HCE) and IOBA-NHC cellular viability was observed by de la Fuente et al. [172] after cytotoxic evaluation of chitosan-HA nanoparticles. The addition of 0.01% (*w*/*w*) HA to a CyA-laden nanomicellar formulation consisting of VitE-TPGS and octylphenoxy poly(ethyleneoxy)ethanol increased rabbit corneal epithelial cell viability from 56–58% to 77% (1.0% (*w*/*v*) CyA) and 93% (0.0001% (*w*/*v*) CyA) [7].

### 3.7. Clinical Applications of HA-Based Ocular Drug Delivery Systems

Due to its remarkable hydrating properties, HA is now the predominant active pharmaceutical ingredient in several commercially available AT formulations (Vismed^®^, Blink^®^ Tears, and Hyalein^®^) and contact lens multi-purpose solution (BioTrue^®^ and Hidro Health) [52,173]. The treatment of moderate-to-severe dry eye disease (DED) has been the primary focus of clinical trials involving HA-based topical eye drop formulations in recent years. A meta-analysis conducted by Yang et al. [157], in which 19 studies with a total of 2078 cases were evaluated, highlighted the superior efficacy of HA-based artificial tear (AT) formulations in comparison to alternative AT and saline solutions in improving DED symptoms. Similarly, Hynnekleiv et al. [174] previously summarised the results of 53 clinical trials involving the use of AT solutions containing 0.1–0.4% HA in the treatment of DED over 3 months. The safety and efficacy of HA-based formulations, demonstrated in some completed clinical trials with published results, will be discussed here.

When compared to autologous serum eye drops derived from the patients’ blood samples, Beck et al. [175] observed that Comfort Shield^®^, an eye drop formulation containing 0.15% HMWHA, exhibited comparable efficacy in treating patients with severe, late-stage dry eye disease in an initial, randomised clinical trial. However, it was noted that, although comparable efficacy was achieved between the two formulations, severe corneal surface irregularities limited the reliability of the corneal fluorescein staining score. Additionally, the authors reported that due to the small size of the cohort (8 in total, 4—control group, 4—Comfort Shield^®^), further clinical trials with a larger cohort would be required to ensure the validity of the results obtained.

In an investigator-masked, randomised clinical assessment, a combination eye drop formulation containing glycerine and erythritol, carboxymethyl cellulose (0.5%), and HA (0.1%) exhibited comparable efficacy to a 0.18% HA formulation, indicated by a statistically similar reduction in Global Ocular Staining Score from baseline after 3 months [176]. Although both formulations were well tolerated, indicated by minimal changes in best-corrected visual acuity and a low incidence of treatment-related adverse effects, there was a significant decrease in the number of reported cases of subjective DED symptoms (burning sensation, itching, etc.) in the combination treatment in comparison to the 0.18% HA formulation. A statistically significant increase in tear film break-up time (TFBT) was reported for patients treated with 0.9% saline solution and 0.1–0.3% sodium hyaluronate-based eye drop formulations (Ocudry^®^) 30 min after instillation in a double-masked, randomised trial [177]. Subjective vision significantly improved post-instillation, with the 0.2% and 0.3% Ocudry^®^ solutions exhibiting a marked improvement from 1 min to 20 min post-instillation. In terms of comfort, Ocudry^®^ 0.3% scored significantly less at 1 min and 5 min post-instillation, attributed to the higher viscosity of the formulation with increasing sodium hyaluronate concentration. However, there was a significant improvement in the subjective comfort score after 20 min from baseline in the Ocudry^®^ 0.3% cohort.

Pinto-Fraga et al. [178] demonstrated the superior efficacy of Visaid^®^, a 0.2% sodium hyaluronate solution, in comparison to a 0.9% saline solution in reducing the Ocular Surface Disease Index (OSDI) score after daily instillation of 3–8 drops in a phase II, double-blind clinical trial. Patients treated with 0.2% Visaid^®^ exhibited a percentage reduction in OSDI index value of −19.5 ± 27.3% from baseline after 1 month, with over 31.2% of patients reporting an OSDI comfort value of greater than 5 points. No significant changes in intraocular pressure (IOP), TFBT, or visual acuity from baseline were reported, thus confirming the tolerability of the formulations. Interestingly, when 0.3% Visaid^®^ was compared to 0.9% saline in a later phase III double-blind clinical trial, a statistically significant increase in TFBT from baseline was recorded (+13.98 ± 26.19%, *p* > 0.05) for patients treated with 0.3% Visaid^®^ [179]. A significant difference in the percentage change in TFBT from baseline was also reported between those treated with 0.3% Visaid^®^ and 0.9% saline control after 1 month. The MW of the sodium hyaluronate within the Visaid^®^ solutions was not reported. Assuming the MW of the solutions is similar, it would have been interesting to investigate whether a similar therapeutic response could be achieved with a lower dose of the 0.3% Visaid^®^ solution due to the concentration-dependent increase in viscosity. A recent clinical evaluation was conducted by Alcon Research to evaluate the safety and efficacy of the dual implantation of a CyPass^®^ Micro-Stent and a commercially available viscoadaptive ophthalmic viscosurgical device (Healon5^®^) in lowering IOP in a cohort with open-angle glaucoma [180]. A total of 83.3% and 73.5% of the patients treated with the CyPass^®^ Micro-Stent and 30 µL of 60 µL of Healon 5, respectively, experienced a reduction in IOP of over 20% from baseline 12 months post-implantation.

## 4. HA Polyelectrolyte Complexes

Self-assembled nanoformulations, particularly those synthesised from biopolyelelectrolytes, such as HA, are gaining considerable scientific interest in the biomedical field. Formed via electrostatic interaction under mild, aqueous formulation conditions, polyelectrolyte complexes (PECs) present a biocompatible, biodegradable drug delivery system with functionalities that can be tailored for a wide variety of pharmaceutical applications during formulation. As reviewed by Cazorla-Luna et al. [181] and Papagiannopoulos [182], PECs have been adapted for use in multiple dosage forms, including films, hydrogels, porous scaffolds for tissue engineering, and nanovesicles for the delivery of genetic material, protein, and small therapeutic molecules. Taking advantage of the pH-dependant stability of PECs, combined with the cell-binding and penetrating capabilities of biopolyelectrolytes, targeted, responsive therapeutic release can be obtained in vivo.

Despite the proven, multi-purpose efficacy of HA in a variety of ocular drug delivery systems (ODDs), and the favourable characteristics of PECs for drug delivery application, limited research has been conducted into the use of HA-based PECs for ocular application. Following a description of colloidally stable PEC formation with respect to intrinsic formulation parameters and complexation thermodynamics, recent publications in the area of HA-based PECs in ODDs will be reviewed.

### 4.1. Polyelectrolyte Complex Assembly Mechanism

Polyelectrolyte complexation is an example of supramolecular self-assembly of polyelectrolyte macromolecules of opposing charge into structurally ordered nano-aggregates in an aqueous solution [183]. In such environments, polyelectrolytes adapt either cationic or anionic surface charges due to the ionisation of the acidic (carboxyl or sulphate) or basic (amine) functional groups within the polyelectrolyte chains. The electrostatically driven self-assembly of polymeric material into submicron-sized particles is advantageous in comparison to more conventional nanoparticle preparation methods. Formed under mild formulation conditions and in the absence of harsh organic solvents, chemical crosslinkers, and surfactants, polyelectrolyte complexes (PECs) exhibit high biocompatibility and are free from residual toxic by-products [184]. Extensive purification post-formation is not required, rendering polyelectrolyte complexation a more cost-effective and environmentally friendly method of nanoparticle preparation.

The supramolecular packing of oppositely charged polyacid and polybase moieties via electrostatic charge compensation and the increase in entropy resulting from counter-ion displacement are the driving forces for the self-assembly of polyelectrolyte chains into a complex coacervate phase [185,186,187]. Polyelectrolyte complexation occurs in three primary steps [188,189]. Firstly, the kinetically driven mutual entanglement and self-assembly of polyelectrolyte chains via Coulomb interactions result in the formation of primary complexes. Thermodynamic stability is attained via intracomplex reconfiguration. Conformational changes or distortions that occurred during primary complexation are corrected via the establishment of secondary supramolecular or non-covalent interactions. Intercomplex aggregation of the structurally organised secondary complexes composes the final step in the complexation process. Two structural models have also been proposed for PEC assembly and conformation [190,191,192]. The ladder-like structural model describes PEC assembly as the ordered packing of opposing polyelectrolyte chains resulting from electrostatic charge compensation via a zip mechanism, followed by conformational adaptation to maximise charge neutralisation. The resulting PECs consist of both ionised, hydrophilic, single-stranded blocks and neutralised, hydrophobic, double-stranded blocks.

Conversely, the “scrambled egg” theory of PEC assembly follows a more chaotic approach. Opposing polyelectrolyte chains randomly entangle and aggregate, resulting in partial charge compensation and the formation of ionised sites within a highly entangled polymeric matrix. Theoretically, the true model of PEC assembly is a combination of both the ladder and “scrambled egg” structural models. The predominating assembly mechanism is contingent on many intrinsic characteristics, such as the MWs, concentrations, and charge densities of the respective polyelectrolytes and the molar charge ratio (MCR), shown below in Equation (1) [193].
(1)$$\mathrm{Molar}\;\mathrm{Charge}\;\mathrm{Ratio}\;\left(\mathrm{MCR}\right)=\frac{\mathrm{Molar}\;\mathrm{units}\;\mathrm{of}\;\mathrm{anionic}\;\left(-\right)\;\mathrm{charge}}{\mathrm{Molar}\;\mathrm{units}\;\mathrm{of}\;\mathrm{cationic}\;\left(+\right)\;\mathrm{charge}}$$

The MCR represents the ratio of moles of anionic to cationic polyionic charges in a PEC system and plays a vital role in both the formation and stabilisation of PEC structures. As shown in Figure 3, the structure of PECs can be sub-divided into three primary categories: water-soluble PECs, colloidally stable PECs, and macroscopically precipitated PEC aggregates. However, for the purpose of this review, colloidally stable PECs will be of particular focus as they are the most suitable structural model for nanoparticle-facilitated drug delivery. The complexation of polyelectrolytes exhibiting non-stoichiometric charge ratios under very high dilution results in the formation of colloidally stable, insoluble PECs. Such complexes do not exhibit thermodynamic stability but are kinetically stabilised via mutual electrostatic repulsion of the residual surface charges, the sign of which is dependent on the polyelectrolyte in excess [194]. Such electrostatic stabilisation is imperative to prevent aggregation in vivo, thus ultimately reducing size-exclusion during intraocular transport and subsequent delocalised, uncontrolled therapeutic release [195].

### 4.2. Thermodynamics of Polyelectrolyte Complexation

The increase in entropy upon disruption of the electrical double layer (EDL) of the individual polyelectrolyte chains and subsequent release of both counterions and solvent molecules into the bulk medium drives complexation [185,197]. The increase in entropy from counterion displacement far exceeds the loss of conformational entropy from the mutual entanglement of the polyelectrolyte chains via supramolecular bond formation. Such favourable entropic contributions, coupled with the decrease in enthalpy resulting from interpolyelectrolyte electrostatic bond formation, ultimately result in a reduction in total Gibb’s free energy of the system and the spontaneous formation of PECs [198].

PECs are not thermodynamically stable. In the absence of kinetic stabilisation via residual surface charges, dispersed PECs will tend towards aggregation to decrease the area at the PEC–bulk medium interface, resulting in macroscopic, thermodynamically stable phase separation [198]. Hydrophobic interactions form between the polyelectrolyte hydrophobic moieties to lower the interfacial energy at the hydrophobe–water molecule interface. The increase in entropy resulting from the displacement and disruption of the hydrogen bonds between the water molecules surrounding the hydrophobic moieties drives hydrophobic interaction further. The surface-to-surface separation distance decreases as a result, thus increasing the formation of attractive Van der Waal’s forces. As per the Derjaguin, Landau, Verwey, and Overbeek (DLVO) theory, describing the theoretical analysis of colloidal stability based on the attractive and repulsive forces acting on charged colloidal particles in solution, the system now exists in a deep potential-energy well due to the predominance of attractive forces at short interparticle distances [188,199]. This results in the irreversible formation of a macroscopic, thermodynamically stable PEC aggregate. To attain colloidal metastability, the energy of the PECs must not exceed that of the electrostatic energy barrier (Δ_int_(G))max, achievable via kinetic stabilisation of the PECs via surface charge.

### 4.3. Applications of HA Polyelectrolyte Complexes in Ocular Drug Delivery

At physiological pH, HA possesses a negative charge due to deprotonation of the carboxylate anion (pKa 3–4) and can readily complex with cationic species [200,201]. Such polycations include CS, chitosan derivatives such as trimethyl chitosan, poly(L-lysine), silk fibroin, dextran derivates, and poly(arginine) [194,202,203,204,205,206,207,208]. Silva et al. [209] synthesised chitosan and HA nanoparticles via electrostatic complexation for use in a topical eye drop solution of erythropoietin. Optimal ZP values and nanoparticle sizes were obtained when the concentration of chitosan and HA of varying MWs were in equal ratios. Significant aggregation was observed when chitosan was present in excess (chitosan:HA ratio—1:0.5), which they attributed to the mixing order of the polyelectrolytes during formulation. The addition of chitosan in excess of HA at this charge ratio may have resulted in the neutralisation of the anionic carboxylate moieties of the initial coacervates before all of the chitosan was introduced to the formulation. Enhanced ex vivo erythropoietin permeation through porcine conjunctiva (0.71 ± 0.04%), sclera (0.52 ± 0.08%), and cornea (0.17 ± 0.04%) were achieved via HA–chitosan complex encapsulation when compared to the percentage ex vivo permeation values obtained for erythropoietin alone (0.52%, 0.35%, and 0.07%). However, an in vitro burst release of erythropoietin, equating to approximately 70% of the erythropoietin loaded, was observed within 30 min. This may prove problematic for treatments designed for topical ocular administration as a faster rate of release shortly after administration may result in increased therapeutic elimination from the pre-corneal area due to nasolacrimal clearance.

The release of 50% of the loading concentration of pentamidine isethionate (PTM), a hydrophilic antiprotozoal and antifungal agent, from HA and poly(arginine)-based PECs was also observed by Carton et al. after 8 h [210]. This was primarily attributed to the hydrophilicity of PTM, resulting in both faster permeation through the HA-poly(arginine) matrix upon release and a higher degree of PEC swelling. Similar in vitro release profiles for the PTM-PEC and free PTM were also recorded, demonstrating that the PECs may not have provided a sufficient diffusion barrier to PTM release when dispersed in the receptor medium. Through ion-mediated diffusion and exchange with the protonated amino groups of timolol maleate™ in 0.9% NaCl solution, Battistini et al. [211] reported an approximate 10% increase in the cumulative release of TM) from sodium hyaluronate (NaHA)-TM ionic complexes over 8 h in comparison to that obtained when water was used as the receptor medium. The NaHA-TM complexes also exhibited superior efficacy in reducing IOP in comparison to a 0.5% TM commercial eye drop (Zopirol^®^) in a rabbit model over 10 h, attributed to the enhanced mucoadhesive capabilities of the NaHA-based formulation. Increased TM bioavailability in rabbit aqueous humour over 4 h, evidenced by a significantly higher area under the curve (607.7 ± 53.4 µg·h/mL) was also observed for the NaHA-TM formulation in comparison to Zopirol^®^ (400.5 ± 12.2 µg·h/mL).

Chaharband et al. [208] designed trimethyl chitosan and HA nanopolyplexes for the intravitreal delivery of vascular endothelial growth factor 2 (VEGF-2)-small interfering RNA (siRNA). Complete retinal penetration of the nanopolyplexes was visible after one-hour post-IVI in a rabbit model, which they attributed to the ability of HA to permeate the blood-retinal barrier and vitreous humour. Retinal distribution was still detected 120 h post-IVI. Laser-induced choroidal neovascularisation was reduced in the nanopolyplex-treated group (46,501 ± 21,739 µm^2^) in comparison to the control group (113,728 ± 32,289 µm^2^) in a murine model.

### 4.4. HA-PECs: Design Considerations and Critical Quality Attributes

Many reviews have discussed the dependence of PEC stability and formation on numerous critical intrinsic and extrinsic factors [63,184,212]. To design an HA-PEC formulation that is both stable under physiological ocular conditions and capable of releasing a drug in a sustained and controlled manner, such attributes must be optimised during formulation method development.

#### 4.4.1. pH

The pH value of the formulation medium has a significant impact on the degree of ionisation of the respective polyelectrolytes within the PECs, thus ultimately affecting the extent of electrostatic bond formation and the separation of PECs into a polymer-rich phase [194,210,213]. A pH value of 7.2 has been reported for the outer nuclear retinal layer, whilst tear fluid exhibits an average pH value of 7.4 [214,215]. Based on its pKa, the carboxylic acid groups of HA should be completely ionised at the aforementioned pH values in the anterior and posterior segments. However, to ensure maximal complexation, the pKa value of the cationic polyelectrolyte should also be taken into consideration. Potentiometric titrations are often conducted to determine the pH value at which both polyelectrolytes exhibit maximal ionisation [216,217]. The extent of ionisation of the PEC formulation at physiological pH should also be determined to ensure that the electrostatic interactions formed are not diminished upon application.

#### 4.4.2. PEC Surface Charge

PECs in which HA is present in excess (n^−^/n^+^ > 1) will preferentially adopt an anionic surface charge. It has been suggested that HA chains will form a peripheral corona, surrounding the charge-neutralised, hydrophobic core, with the electrostatically charged, hydrophilic carboxylate anions residing at the interface of the PEC and the aqueous media [218]. PECs bearing an anionic surface charge are desirable for particular ocular applications. For example, high therapeutic bioavailability within the inner retinal tissues post-intravitreal-administration can be attained via encapsulation within an anionic PEC, primarily due to increased vitreal permeation resulting from the electrostatic repulsion between HA and the negatively charged vitreal glycosaminoglycan moieties [168]. Anionic HA-PECs would also pose a lower risk of cytotoxicity in comparison to cationic PECs. This is also attributed to increased electrostatic repulsion between the residual carboxylate anions and both the phosphate head groups and anionic glycoproteins within the phospholipid bilayer of the cell membrane [219].

#### 4.4.3. Ionic Strength and Stability

When dispersed in a medium of high ionic strength, the ingression of ions into the PEC matrices can effectively screen the electrostatic repulsive forces between kinetically stabilised, charged, colloidal PECs. In comparison to Figure 4a, the reduction in the thickness of the EDL, i.e., the Debye length (λD) (shown in Figure 4b), due to high surface electrostatic charge screening, results in counterion cloud compaction. The increase in energy (+H) required to displace ions from this compact counterion cloud is far greater than the reduction in energy (−H) from the formation of polyelectrolyte ion pairs. As such, PEC formation under these conditions is endothermic. Such processes lead to a reduction in interparticle repulsion, allowing for increased overlap between the EDLs of PECs in close proximity to one another. As per the DLVO theory, at short interparticle distances, Van der Waals forces of attraction tend to predominate. This, coupled with the elimination of the electrostatic energy barrier in high-ionic-strength conditions, can result in irreversible, thermodynamically favourable aggregation [220,221,222,223]. Above a critical ionic strength value, PEC nucleation is drastically hindered due to extensive electrostatic charge screening, resulting in the re-dissolution of the individual polyelectrolytes [220,221,224]. The average electrolyte concentration of human tear fluid, based on the concentration of sodium, potassium, chloride, and bicarbonate ions, is approximately 0.3 M (300 mmol/L) [225,226,227]. Therefore, it is imperative that the stability of PECs in a medium with an ionic strength mimicking that of physiological tear fluid be evaluated during the formulation design stage. Premature PEC degradation in vivo could result in delocalised drug absorption away from the desired target site, thus ultimately reducing the bioavailability of the formulation. As a means of preserving the stability of chitosan and HA-based PECs, Wu et al. [207] investigated the addition of zinc (II) salts to the PECs during formation. This may be a suitable option to enhance stability under physiological conditions as zinc (II) can readily form bridging coordinate bonds with electron-rich donor groups on neighbouring HA chains.

#### 4.4.4. The Control of Therapeutic Release from PECs

As highlighted in Section 4.3, therapeutic-laden PECs tend to exhibit fast in vitro release kinetics when placed in an aqueous receptor medium. Hydrophilic drugs will preferentially associate with the hydrophilic corona of the PECs. Cationic drugs can also complex to HA-PECs with non-stoichiometric MCRs (n^−^/n^+^ > 1) directly, due to the presence of residual carboxylate anions. However, the absence of a highly complexed polymeric diffusion barrier at the PEC surface, and the establishment of an osmotic gradient between the aqueous donor phase and the receptor phase, could result in accelerated therapeutic release. Fast intraocular release is unfavourable, particularly for sustained-release ocular drug delivery systems designed to minimise frequent, invasive administration. Additionally, similar to the issues arising from PEC instability in high-ionic-strength conditions, premature release, coupled with the limited migration of therapeutics through the anatomical ocular barriers, will significantly reduce the bioavailability of the therapeutic at the target site, particularly if that target site lies within the posterior segment. Drug molecules with a higher degree of lipophilicity tend to associate with the charge-neutralised hydrophobic core via supermolecular interactions, and would assumingly exhibit slower rates of release due to incompatibility with the external aqueous phase. However, the initial entrapment of lipophilic molecules may prove problematic due to limited solubility in the aqueous formulation media. To control the release of hydrophilic drugs, and to ensure the maximal complexation of lipophilic drugs within the PEC core, the chemical conjugation of drugs to the polymeric backbone of HA via a cleavable linkage may be required. Although it overcomes the pitfalls of physical therapeutic absorption, chemical conjugation may introduce toxicity via residual solvent and coupling agents, thus ultimately affecting the biocompatibility of the PEC formulation [51,118,200,228].

#### 4.4.5. The Influence of HA Concentration and MW

As discussed in Section 2.3, both the concentration and MW of HA have a considerable impact on the viscosity of the final formulation, an attribute that must be carefully optimised to ensure maximal ocular retention whilst minimising potential adverse effects such as ocular irritation and blurred vision. Regarding the administration of drug-laden HA-PECs via IVI, the intrinsic viscosity of the formulation should not exceed that of the vitreous humour (0.1–0.85 Pas) [229]. However, an inverse correlation exists between viscosity and injection volume. Therefore, the formulation should exhibit sufficient viscosity to ensure a minimal volume of the formulation is required for injection, thus preventing potential increases in IOP resulting from the injection of a large volume into the small, fixed vitreal volume of 4 mL [73,230].

With HA-PECs, the effect of HA concentration and MW on PEC stability must also be evaluated. With increasing concentration and MW, an increased number of counterions are released into the EDL surrounding the newly formed PECs. The dual reduction in Debye length and electrostatic stabilisation with increasing counterion concentration promotes the secondary aggregation of primary PECs, leading to the formation of PECs with unfavourable size characteristics. Additionally, similar to what was observed by Schatz et al. [231] when developing chitosan and dextran sulphate PECs, the MW and flexibility of the polyelectrolyte in excess play a fundamental role in PEC colloidal stability. For the formation of anionic PECs (HA in excess), if the MW of the HA used is less than that of the cationic polyelectrolyte, flocculation may occur due to complete polyelectrolyte–polyelectrolyte ion-pairing via the conformational adaptation of HA. However, due to the rigidity of the 1-4-glycosidic bonds, HA may resist adaptation in favour of ion-pairing, allowing the cationic polyelectrolyte of higher MW to behave as a host molecule whilst attaining residual anionic charges for electrostatic stabilisation.

## 5. Conclusions

In summary, this review highlights the suitability of HA, based on its favourable physicochemical and biological properties, for use as an excipient in ocular drug delivery systems. The mucoadhesive and viscoelastic properties of HA, combined with its increased cellular uptake via CD44-mediated endocytosis, allow for localised, targeted delivery of therapeutics within the anterior and posterior segments in a sustained manner. Due to its inherent hydrophilicity and biocompatibility, HA has shown great success in providing maximal comfort whilst preserving cellular homeostasis and avoiding potentially harmful adverse effects upon application. However, despite the extensive use of HA-based formulations in ophthalmology and ocular drug delivery, limited research has been conducted into the use of HA-PECs for ocular applications, particularly for posterior segment delivery. There is scope to formulate a system that combines the favourable physicochemical and biological attributes of HA with highly biocompatible PECs that offer similar advantages to those of conventional nanoparticle formulations, whilst avoiding the use of harsh formulation conditions. However, to achieve maximal efficacy with posterior segment delivery, the PEC formulation must remain stable within the intraocular environment whilst permeating through the various anatomical diffusion barriers. Due to the fragility of the interpolyelectrolyte electrostatic interactions, particularly in high-ionic-strength conditions and fluctuating pH, further modification of HA-PECs via chemical conjugation, salt-based stabilisers, or lipoidal coatings may be required for maximal stability and controlled release at the target site. However, with careful optimisation of the critical quality attributes and formulation parameters, HA-based PECs yielding optimal physicochemical parameters have been successfully formulated in the selected examples, which opens the possibility for the development of HA-based PECs for a diverse range of ophthalmic applications.

## Figures and Tables

**Figure 1 pharmaceutics-14-01479-f001:**
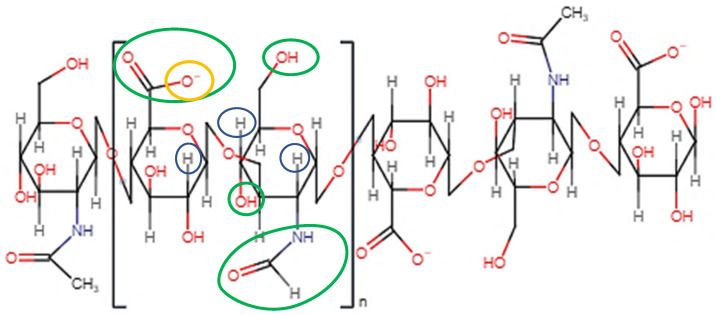
A schematic of the two-fold helical structure of HA. Colour coding is used to indicate the axial hydrogen atoms (blue), the equatorial side chains (green), and the anionic carboxylate group (orange) within the repeating disaccharide unit.

**Figure 2 pharmaceutics-14-01479-f002:**
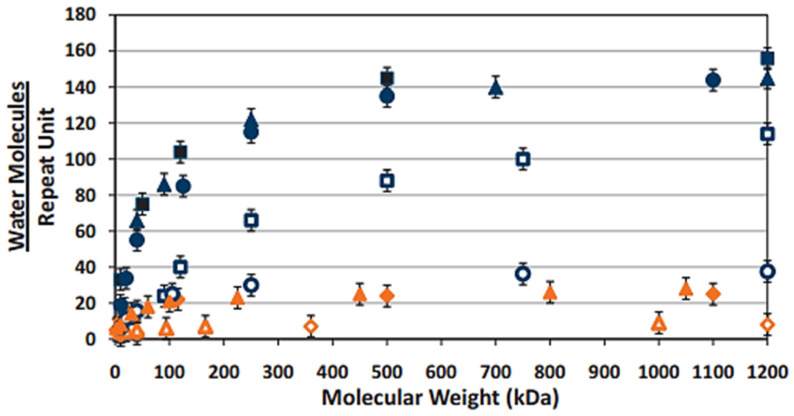
Water retention per polymer repeat unit of various polyelectrolyte polysaccharides and acrylics. The water retention of HA, (

) exhibiting superior water retention in comparison to the other polyelectrolyte materials, increases as a function of MW. (

—dextran sodium sulfate, 

—hydroxypropyl methylcellulose, 
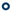
—dextran, 
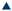
—carboxymethylcellulose, 

—poly(acrylic acid), 

—poly(methacrylic acid), 
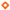
—polyvinyl alcohol, 

—polyvinyl pyrrilidone). (Reproduced with permission from Ref. [122]. 2014, White, C.J., Thomas, C.R. and Byrne, M.E. “Bringing comfort to the masses: A novel evaluation of comfort agent solution properties”; published by Contact Lens and Anterior Eye.

**Figure 3 pharmaceutics-14-01479-f003:**
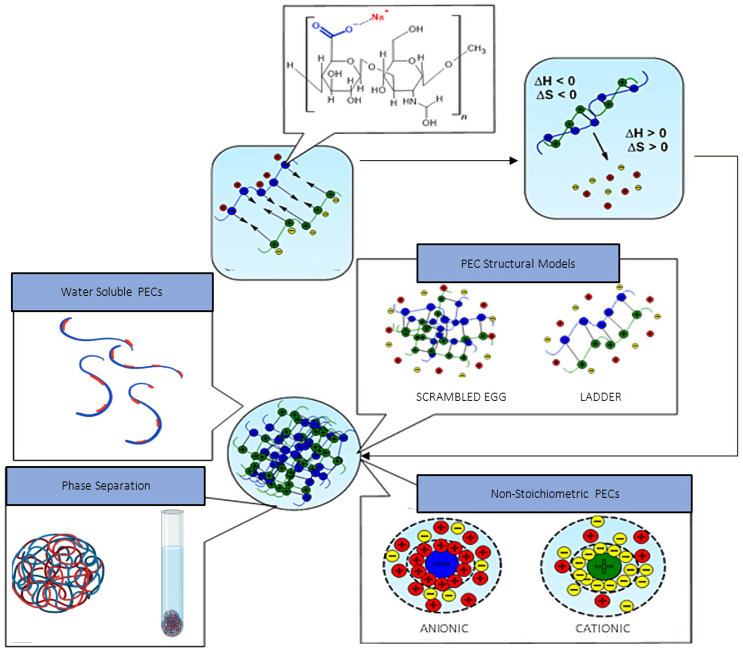
An overview of the complexation between HA and a cationic polyelectrolyte with the resulting physiochemical attribute-dependent structural models (blue—carboxylate anion, red—cationic counterion, green—cationic polyelectrolyte, yellow—anionic counterion; ΔH—change in enthalpy, ΔS—change in entropy) (Created using Biorender.com [196]).

**Figure 4 pharmaceutics-14-01479-f004:**
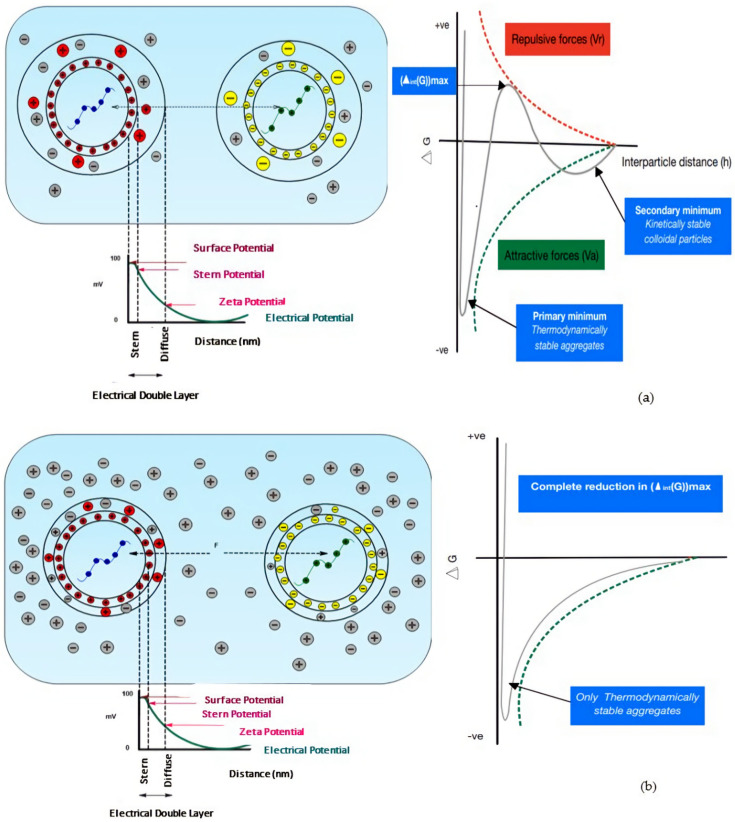
The complexation of two polyelectrolytes of opposing charges under low–(**a**) and high–(**b**) ionic–strength conditions. The effect of both low and high ionic strength on colloidal stability is also represented by the energy vs. interparticulate distance graph drawn per DLVO theory (Red (+)—cations within the electrical double layer (EDL) of HA that are strongly attracted to the anionic carboxylate groups. Yellow (−)—anions within the EDL of the cationic polyelectrolyte that are strongly attracted to the cationic groups. Grey (+/−)—counterions within and outside the EDL of the polyelectrolytes that are more weakly attracted to the surface charges).

**Table 1 pharmaceutics-14-01479-t001:** The physicochemical state of HA from pH 1 to 14 [65,66,67,68].

pH	Ionisation	Stability	Viscosity
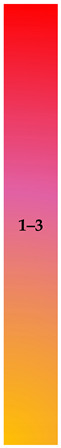	The **carboxylate anion** primarily exists in its protonated state (COOH).Protonation of the **-NH- group** of the acetylamine groups may also occur at a pH < 1.6. 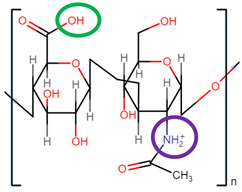	At a pH of <2, cleavage of the β-1-3 and β-1-4 glycosidic bonds via acid hydrolysis occurs. This results in the reformation of the individual monosaccharide units coupled with a decrease in molecular mass. 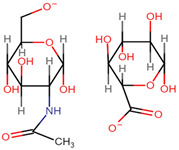	A significant decrease in HA viscosity occurs within this pH range. pH-induced degradation of the individual HA chains reduces both the rigidity and the extent of interchain entanglement. Suppression of electrostatic repulsion between the COO^-^ anions contributes to the formation of a more densely compact gel state. Below 1.6, a gel-sol transition occurs due to acetylamino group protonation.
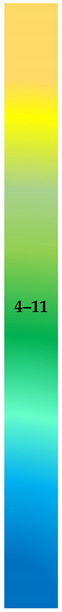	The **carboxylate anion** exists in its deprotonated state (COO^−^). Electrostatic repulsion between COO- anions allows for hydrophilic domain expansion. Protonation of the **-NH- groups** is at a minimum. 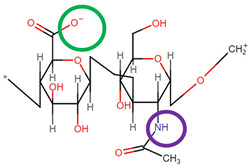	The stability of the tertiary β sheet structure is enhanced by the formation of an intermolecular hydrogen bond between thehydroxyl and ether groups 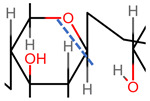 and carboxylate anions and acetylamine groups. 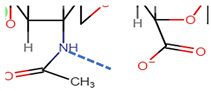 Water molecules can also indirectly facilitate the formation of these hydrogen bonds by behaving as bridging molecules.	HA exhibits both viscoelastic and pseudoplastic flow behaviour. Increasing shear rate disrupts hydrogen bonding and hydrophobic interactions, resulting in increased chain flexibility and network degradation. This allows the individual chains to align in the direction of the applied flow, leading to a temporary decrease in viscosity. However, HA is also non-thixotropic. Reformation of the network occurs over time upon removal of the shear stress.
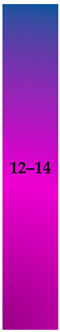	At pH > 12, the **COO- anions** remain in their deprotonated state. The **hydroxyl (OH) atoms** also exist as alkoxyl anions due to the removal of the hydrogen atoms via excess hydroxide ions. 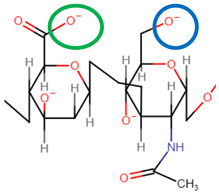	Base-catalysed hydrolysis of the β-1-3 and β-1-4 glycosidic bonds occurs. Due to the deprotonation of the hydroxyl groups, the hydrogen bonds responsible for the stabilisation of the tertiary network begin to degrade. 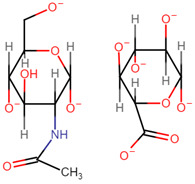	Deformation of the intermolecular hydrogen bonds reduces the rigidity of the backbones of the HA chains. This results in the degradation of the entangled network, coupled with an increase in chain flexibility and mobility.

**Table 2 pharmaceutics-14-01479-t002:** A synopsis of HA-based nanoformulations in ocular drug delivery.

Formulation	Preparation Method	Excipients	Function	HA MW (kDa)	Desired Effect	Therapeutic	Key Findings	Further Investigation	Ref.
**Nanoparticles**	Ionotropic Gelation	Chitosan (CS) and tripolyphoshapte (TPP)	Coating agent	200	Improved cellular targeting via receptor-mediated internalisation.	Dexamethasone sodium phosphate (DEX-SP)	Increase in size from 305 ± 14 nm to 386 ± 13 nm and reversal of ZP values from high positive to high negative with HA coating. Coating with HA decreased initial DEX -SP in vitro burst release by approximately 15%. Similar release profiles were obtained between the 1 and 12 h time points.	In vitro/ex vivo mucoadhesion In vitro cytotoxicity analysis In vitro CD44- binding and cellular uptake analysis (study conducted by Matha et al. [156] to investigate HA-CD44 mediated cellular uptake).	[157]
**Nanofibres**	Electrospinning	Polyvinylpyrrolidone	Excipient	600–1100	Enhanced ocular bioavailability after conjunctival application.	Ferulic acid (FA) and ε-poly(L-lysine) (ε-PLL)	Crosslinking the HA nanofibres with ε-PLL formed electrostatically cross-linked nanofibre-laden inserts (blank inserts only). Blank and FA-loaded inserts preserved chorioallantoic membrane integrity (embryonated hen’s egg).	Ex vivo conjunctival permeation and retention studies.	[158]
**Nanoparticles**	Desolvation	Human serum albumin	Surface ligand	120	Targeted retinal delivery via active CD44 targeting.	Connexin 43 mimetic peptide (Cx43-MP)	The HA-coated nanoparticles exhibited enhanced ex vivo retinal penetration into the outer nuclear layer and retinal pigment epithelium in comparison to the uncoated control group 4 h post-incubation due to high CD44 ligand interaction. 1.2- and 1.5-fold increase in fluorescence intensity in neural retina and retinal pigment epithelium/choroid with HA-coated particles in comparison to the uncoated group. HA-modification exhibited superior ARPE-19 in vitro internalisation in comparison to control.	In vivo permeation studies. In vivo Cx43-MP quantification to determine whether nanoparticle encapsulation prolongs the half-life of Cx43-MP in the vitreal injection site after injection.	[159]
**Lipid–polymer hybrid nanoparticles (LPHNP)**	Ionotropic gelation and thin-film hydration	CS, TPP, Lipoid E 80, cholesterol and DPPE	Surface Ligand	10	Enhanced corneal retention and permeability.	Moxifloxacin hydrochloride (MCF.HCL)	The apparent permeability coefficient of the HA-coated LPHNPs through excised rabbit cornea was 3.29- and 1.69-fold higher than those of the MXF commercial product and the uncoated LPHNP formulation. The area under the curve of the HA-coated LPHNPs between 0 and 6 h was 6.74 and 2.56 times higher than that of the commercial MXF product and the control chitosan nanoparticle formulation, respectively. Fluorescent-labelled HA-coated LPHNPs exhibited stronger fluorescence within the corneal and conjunctival tissues of a rabbit model after ex vivo fluorescent imaging analysis.	Although the enhanced precorneal retention of the HA-coated LPHNP formulation was attributed to the mucoadhesive capabilities of HA, no mucoadhesion studies were conducted. In vitro stability studies to evaluate the stability of the HA-coated LPHNPs in comparison to the uncoated LPHNP and control nanoparticle formulations.	[160]
**Nanogels**	Physical crosslinking	ε-PLL	Excipient	200/700/1200	General wound healing capabilities.	Berberine	Due to macrogelation and polydispersity index values above 0.3, nanogels formulated with 700 and 1200 kDa HA or HA concentrations above 2 mg/mL were excluded from further characterisation studies. In vitro berberine release was sustained for 24 h after an initial burst release of approximately 50% of the total loaded berberine within 45 min. Blank nanogels exhibited greater efficacy in promoting in vitro wound healing over 48 h in comparison to the berberine-loaded nanogels, which they attributed to the decreased availability of HA as a result of electrostatic interaction with berberine.	A rheological assessment of optimised nanogels. These nanogels were prepared for general wound healing purposes. If designed for corneal wound healing, nanogels exhibiting high viscosity may be beneficial for increasing the retention time and pharmacological profile of the formulation.	[161]
**Nanoparticles**	Self-assembly	Gelatin	Surface ligand	Not Listed	Mucoadhesion to the corneal–conjunctival interface.	Epigallocatechin gallate (EGCG)	HA-coated gelatin–EGCG nanoparticles (GEH) exhibited greater accumulation in human corneal epithelial cells in comparison to control gelatin–EGCG nanoparticles. 100 min after in vivo administration in a murine model, the GEH formulation demonstrated longer retention on the ocular surface in comparison to both the control nanoparticles and an EGCG solution. After twice-daily administrations for three weeks, the GEH formulation restored normal corneal architecture following benzalkonium chloride-induced dry eye syndrome (DES) in a rabbit model.	A study to evaluate the effect of HA MW on the mucoadhesive capabilities and ocular surface retention times of the GEH formulation.	[162]
**Nanomicelles**	Self-assembly	D-α-Tocopherolpolyethylene glycol succinate (VitE-TPGS) and octylphenoxy poly(ethyleneoxy)ethanol	Excipient	1650	Mucoadhesion to the corneal surface and prevention of corneal damage caused by surfactant excipients.	Cyclosporine A (CyA)	The addition of HA to the micellar formulation significantly decreased the rate of cyclosporine A elimination from the corneal surface in a rabbit model, as evidenced by an elimination rate constant that was almost 4 times smaller than that of the commercial product (Ikervis^®^) and a five-fold increase in CyA half-life.	Additional in vivo efficacy studies to determine the effective dose of CyA required to treat DES. The dual efficacy of the nanomicellar formulation in delivering CyA for DES treatment and minimising corneal surface abnormalities via HA-mediated re-epithelisation and wound healing.	[7]
**Nanoparticles**	Ionotropic gelation	CS and TPP	Surface ligand	Not Listed	Mucoadhesion to the corneal and conjunctival surface.	Latanoprost	Treatment with HA-CS-latanoprost link nanoparticles led to a significant reduction in daily IOP measurements (27.3 ± 2.2% reduction) in comparison to a 0.005% latanoprost eye drop and Xalatan^®^ (0.005%) over a three-day treatment period in a rabbit model.	In vivo cytotoxicity and ocular tolerability analysis. In vivo latanoprost release and quantification in rabbit tear fluid and/or the mucus layer.	[163]
**Nanomicelles**	Co-solvent evaporation	HA-ethylenediamine-hexadecyl group derivatives	Excipient	7.3	Mucoadhesion and enhanced retention time on the corneal surface.	Imatinib	HA micellar decoration with polyethylene glycol (PEG) and L-carnitine (CRN) improved transcorneal permeation in an in vitro human corneal epithelial cell model and an ex vivo bovine cornea model. Corneal permeation coefficients increased by 10.5 (free HA), 20.5 (HA-PEG), and 16.5 (HA-CRN) times in comparison to an imatinib suspension (0.5 mg/mL).	In vivo imatinib pharmacokinetic analysis.	[164]
**Nanoparticles**	Turkevich method	Gold	Coating agent	5	Improved intraocular mobility and targetability.	N/A	Chemical conjugation of thiolated HA to the surface of the gold nanoparticles allowed for enhanced diffusion through retinal explants (from the ganglion cell layer to the photoreceptor layer) in comparison to the uncoated nanoparticles.	Long-term in vitro analysis of antiangiogenic and antioxidant effects of gold nanoparticles. Physicochemical analysis of therapeutic-laden, HA-coated gold nanoparticles.	[165]
**Nanoparticle**	Desolvation	Bovine serum albumin	Coating agent	1400	Enhanced binding to CD44 receptors expressed on RPE-19 cells and minimisation of diabetic vascular adverse effects.	Apatinib	In comparison to uncoated nanoparticles (3.55 ± 0.81 fluorescence intensity), the HA-coated nanoparticles exhibited greater retinal accumulation (12.28 ± 1.39 fluorescence intensity) 5 h post-topical administration in a murine diabetic retinopathy model. The viscosity of the experimental mucin increased from 10.5 ± 0.2 cP to 28.42 ± 1.25 cP upon mixing with the HA-coated albumin nanoparticles. The ZP values of the coated nanoparticles also decreased from −37.3 ± 1.8 mV to −11.9 ± 0.8 mV upon incubation with mucin.	Evaluation of increased apatinib-laden HA-coated nanoparticle cellular uptake resulting from CD44-mediated endocytosis.	[166]
**Nanogels**	Mixing of amphiphilic polymers at various mass ratios	PLGA-PEG-PLGA triblock copolymer	Therapeutic	800	Extended and controlled release of HA via non-covalent modification.	Fluorescein isothiocyanate-HA	5–30 kDa and 30–70 kDa PLL chains sustained the release of HA from the nanogels up to 30 days via counterion-mediated overlap of the anionic HA chains and cationic PLL. This prevented excessive swelling of the electrostatic complexes and allowed for a more gradual in vitro release of HA.	In vivo HA release and pharmacodynamic study to ensure that sustained HA release and optical transparency can be obtained in physiological conditions.	[167]

## Data Availability

Not applicable.

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
