# Peer review of "Hyaluronic Acid: Its Versatile Use in Ocular Drug Delivery with a Specific Focus on Hyaluronic Acid-Based Polyelectrolyte Complexes"

_pharmaceutics, 2022, doi:10.3390/pharmaceutics14071479_

Round 1

Reviewer 1 Report

The manuscript provides a comprehensive review of HA as an excipient in ocular drug delivery after summarizing the physicochemical and biological properties of HA. Meanwhile, the research on HA-based polyelectrolyte complexes is focused.

Hopefully the authors can reconsider the title of the article to better match the content of the manuscript.

Please check for duplicates in the bibliography.

Author Response

I would like to take this opportunity to thank you for reviewing our  manuscript. The time and effort that you  have dedicated to revising our submitted manuscript and providing insightful feedback are greatly appreciated.

Point 1: Hopefully the authors can reconsider the title of the article to better match the content of the manuscript.

Response 1: Thank you for this suggestion. We have taken your comments on board and the title of the manuscript has been updated to “Hyaluronic Acid: Its Versatile Use in Ocular Drug Delivery with a Specific Focus on Hyaluronic Acid-Based Polyelectrolyte Complexes” to capture the main body of the manuscript, the use of hyaluronic acid (HA) as a versatile excipient in ocular drug delivery, and highlight the specific focus of the manuscript, HA-based polyelectrolyte complexes for ocular application.

Point 2: Please check for duplicates in the bibliography.

Response 2: Thank you for this correction. The bibliography has now been corrected and duplicate references have been removed.

Reviewer 2 Report

 The scientific quality of the manuscript is insufficient for publication in its current form. Specific questions and points requiring attention are itemized below.

Reviewer comments: It is still difficult to find the novelty of the work concerning what has already been published. A literature review is required. What is the difference between what is published with what the authors want to publish? It is not clear. The authors must describe these differences.

Reviewer comments: What is the objective?

Reviewer comments: The number of new publications in the field is high and growing day by day. But it is also true that the number of (good) reviews in the area is equally increasing. Hence, I believe new reviews should be focused on the recent advances while making use of the efforts from previous reviews to substantiate the knowledge in the field.

Reviewer comments: The structure of the review is confusing. Firstly, the authors describe the use of hyaluronic acid for ocular drug delivery applications. After, unexpectedly began to write the basis of polyelectrolyte complexes, which is different from what was initially described. The authors must organize this mess.

Reviewer comments: (line 37, page 1) “In the past decade, many novel therapeutic delivery systems for the treatment of ocular diseases affecting both the anterior and posterior segments have been studied”. The authors must describe examples of these type of novel therapeutic delivery systems.

Reviewer comments: The quality of the figures must be improved.

Reviewer comments: In the introduction section: explain the advantages and disadvantages of hyaluronic acid for ocular drug delivery. Also, describe other polymers used.

Reviewer comments: In the introduction sections, authors must describe the main physicochemical, mechanical, and biological properties that must possess the polymeric systems for ocular drug delivery applications.

Reviewer comments: The authors must do a section with the biomaterials produced from hyaluronic acid, for example., hydrogels, films, drop eyes etc etc.

Reviewer comments: The sectionIn Situ Forming Hydrogel Systems” did not show any reference…. why?. The authors can cite: https://doi.org/10.1007/978-981-16-7152-4_3

Author Response

I would like to take this opportunity to thank you for reviewing our manuscript. The time and effort that you have dedicated to revising our submitted manuscript and providing insightful suggestions and feedback are greatly appreciated.

Reviewer comments (1): It is still difficult to find the novelty of the work concerning what has already been published. A literature review is required. What is the difference between what is published with what the authors want to publish? It is not clear. The authors must describe these differences.

Reviewer comments (2): What is the objective?

Responses 1 and 2: Upon review of the initial manuscript, we agree that the novelty of the work in comparison to what has already been published in literature required clarification. Therefore, the abstract has been modified from lines 27-36 (copied below) to further highlight how this review manuscript is bridging the identified gap in the literature, a review that encompasses the proven efficacy of HA in ocular drug delivery, a description of polyelectrolyte complexation mechanisms and thermodynamics and a critical analysis of HA-based polyelectrolyte complexes for ocular drug delivery.

                        Lines 27-36

“The pivotal focus of this review is a discussion of the formation of HA-based nanoparticles via polyelectrolyte complexation, a mild method of preparation driven primarily by electrostatic interaction between opposing polyelectrolytes. To the best of our knowledge, despite the growing number of publications centred around the development of HA-based polyelectrolyte complexes (HA-PECs) for ocular drug delivery, no review article has been published in this area, This review aims to bridge the identified gap in the literature by (1) reviewing recent advances in the area of HA-PECs for anterior and posterior ODD, (2) describing the mechanism and thermodynamics of polyelectrolyte complexation, and (3) critically evaluating the intrinsic and extrinsic formulation parameters that must be considered when designing HA-PECs for ocular application”.

Reviewer comments (3): The number of new publications in the field is high and growing day by day. But it is also true that the number of (good) reviews in the area is equally increasing. Hence, I believe new reviews should be focused on the recent advances while making use of the efforts from previous reviews to substantiate the knowledge in the field.

Response 3: The predominant focus of this review is the recent advances made in the area of HA-based polyelectrolyte complexes for ocular drug delivery and parameters that must be considered when designing such formulations for ocular applications. To substantiate this, reviews and research articles centred around the stimuli-responsive physicochemical attributes of HA, its established use in ocular drug delivery in eye drops, injectable hydrogels, contact lenses etc. and polyelectrolyte complexation have been referenced throughout as follows:

Lines 106-108

“A thorough understanding of polysaccharidic and protein biopolymers such as chitosan, alginate, albumin, dextran and gelatin can be seen in the reviews of Irimia et al. and Allyn et al. [39, 40].”

Lines 374-376

“Many reviews have discussed the development of HA biomaterials, such as injectable hydrogels, nanofibrous membranes and films and vitreous substitutes [59, 60].”

Lines 425-427

“Previous reviews have reported on the use of HA in therapeutic-laden eye drop formulations to both increase ocular residence time and enhance the therapeutic bioavailability and aqueous solubility of lipophilic therapeutics [52, 118].”

Lines 799-802

“As reviewed by Cazorla-Luna et al. [181] and Papagiannopoulos [182], PECs have been adapted for use in multiple dosage forms, including films, hydrogels, porous scaffolds for tissue engineering and nanovesicles for the delivery of genetic material, protein and small therapeutic molecule delivery.”

Lines 963-964

“Many reviews have discussed the dependence of PEC stability and formation on numerous critical intrinsic and extrinsic factors [63, 184, 212].”

Reviewer comments (4): The structure of the review is confusing. Firstly, the authors describe the use of hyaluronic acid for ocular drug delivery applications. After, unexpectedly began to write the basis of polyelectrolyte complexes, which is different from what was initially described. The authors must organize this mess.

Response 4: Upon review, we agree with the reviewers' comments around structure and found that the transition from describing the use of HA in ocular drug delivery to a discussion on polyelectrolyte complexation mechanisms was abrupt and may be misleading in comparison to what was initially described in the abstract. Therefore, we have added the following paragraph to Section 4 (page 30, lines 794 to 810) to bridge the gap between the sections and explain why polyelectrolyte complexation is being discussed. A description of the mechanism of polyelectrolyte complexation is followed by a review of recent advances in the field of HA-based polyelectrolyte complexes for ocular drug delivery application.

Lines 794-810

“Self-assembled nanoformulations, particularly those synthesised from biopolyelelectrolytes, such as HA, are gaining considerable scientific interest in the biomedical field. Formed via electrostatic interaction under mild, aqueous formulation conditions, polyelectrolyte complexes (PECs) present a biocompatible, biodegradable drug delivery system with functionalities that can be tailored for a wide variety of pharmaceutical applications during formulation. As reviewed by Cazorla-Luna et al. [180] and Papagiannopoulos [181], PECs have been adapted for use in multiple dosage forms, including films, hydrogels, porous scaffolds for tissue engineering and nanovesicles for the delivery of genetic material, protein and small therapeutic molecule delivery. Taking advantage of the pH-dependant stability of PECs, combined with the cell binding and penetrating capabilities of biopolyelectrolytes, targeted, responsive therapeutic release can be obtained in vivo.

Despite the proven, multi-purpose efficacy of HA in a variety of ocular drug delivery systems (ODDs), and the favourable characteristics of PECs for drug delivery application, limited research has been conducted into the use of HA-based PECs for ocular application. Following a description of colloidally stable PEC formation with respect to intrinsic formulation parameters and complexation thermodynamics, recent publications in the area of HA-based PECs in ODDs will be reviewed.”

Reviewer comments (5): (line 37, page 1) “In the past decade, many novel therapeutic delivery systems for the treatment of ocular diseases affecting both the anterior and posterior segments have been studied”. The authors must describe examples of these type of novel therapeutic delivery systems.

Response 5: We would like to thank the reviewer for this suggestion. We have now modified lines 43 to 50 to include references to studies and reviews detailing the development of novel anterior and posterior segment treatments. Also, as per your earlier advice regarding reviews, we have directed to the reader a review by Gote et al. [1] rather than going into exhaustive detail in the introduction section.

Lines 43-50

“Recent advances made in the development of novel therapeutic delivery systems for the treatment of ocular diseases affecting both the anterior and posterior segments can be seen in the review of Gote et al. [1]. Such delivery systems include, but are not limited to, thermosensitive in situ gelling systems for enhanced corneal adhesion [2, 3], nanoformulations for enhanced therapeutic solubility, targetability and intraocular migration and retention [4–7], modified contact lenses for dual vision correction and therapeutic delivery [8–12], and bioerodible implants for the sustained, intravitreal delivery of anti-vascular endothelial growth factors [13, 14].”

Reviewer comments (6): The quality of the figures must be improved.

Response 6: Upon revision, we recognise that text in some of the images was difficult to read at 100% magnification. Therefore, we have modified the size and sharpness of the images (particularly Figure 4) to improve the overall quality of the image.

Reviewer comments (7): In the introduction section: explain the advantages and disadvantages of hyaluronic acid for ocular drug delivery. Also, describe other polymers used.

Reviewer comments (8): In the introduction sections, authors must describe the main physicochemical, mechanical, and biological properties that must possess the polymeric systems for ocular drug delivery applications.

Responses 7 and 8: Thank you for the suggestions above. We agree with the proposed additions and have modified lines 101 to 123 (copied below) to highlight the advantages and disadvantages of hyaluronic for ocular drug delivery. The physiochemical characteristics of polymers that are suitable for drug delivery applications have also been listed, with reference to recent literature reviews regarding the use of natural polymers in drug delivery systems.

Lines 101-123

“Polymeric ODDS such as nanoformulations, hydrogels and biological stimuli-responsive systems have been developed to counteract the pitfalls of conventional ophthalmic formulations. Both synthetic and biological polymers have been used in the development of ODDs, with synthetic polymers exhibiting increased mechanical strength and stability in vivo whereas the latter has the added advantages of biomimetic cell receptor targeting and enhanced biocompatibility resulting from fast in vivo enzymatic degradation. A thorough understanding of polysaccharidic and protein biopolymers such as chitosan, alginate, albumin, dextran and gelatin can be seen in the reviews of Irimia et al. and Allyn et al. [39, 40]. For the purpose of this review, hyaluronic acid (HA), a naturally occurring, high molecular weight (MW) glycosaminoglycan biopolymer, will be discussed in detail. HA has become a popular excipient in ocular drug delivery formulations as it possesses favourable characteristics such as pseudoplasticity, biocompatibility, water retention capacity and biodegradability. Its tunable viscosity and mucoadhesive properties are favourable for prolonging the precorneal residence time of various topical formulations. Its inherent hydrophilicity, attributed to the sequestration of water molecules within its network structure upon dispersion in aqueous media, allows for greater nanoformulation and therapeutic stability in vivo [41, 42]. Targeted delivery in vivo is also achievable due to CD44 receptor-mediated endocytosis. Such prolonged and targeted therapeutic delivery at the target site is highly beneficial for reducing the frequency of administration and improving patient compliance, particularly for diseases affecting the posterior segment that are treated via invasive surgical procedures. However, HA undergoes rapid enzymatic degradation in vivo and may require physical or chemical modification to achieve therapeutically relevant stability in vivo [40].”

Reviewer comments (9): The authors must do a section with the biomaterials produced from hyaluronic acid, for example., hydrogels, films, drop eyes etc etc.

Response 9: Thank you for the suggestion. The title of Section 3 (page 11) has now been updated to “3. Hyaluronic Acid Biomaterials for Ocular Drug Delivery” to highlight the HA-based biomaterials discussed in this section. We agree that the use of HA-based biomaterials is not limited to the examples detailed in this section and although HA-based membranes and films have not been included in this manuscript, we have referenced reviews [59] and [60] and research articles that discuss such biomaterials in much greater detail. Lines 373 to 379 (copied below) include the references for the aforementioned literature.

Lines 373-379

“Due to its tunable physicochemical properties and inherent biocompatibility, the use of HA as an ophthalmic excipient has gained significant traction in recent years [51]. Many reviews have discussed the development of HA biomaterials, such as injectable hydrogels, nanofibrous membranes and films and vitreous substitutes [59, 60]. Similarly, recent studies have also demonstrated the efficacy in enhancing wound healing, protecting corneal epithelium cells, and attaining sufficient intraocular ocular lubrication and pressure during vitro-retinal and cataract surgery [3, 42, 79, 106–108].”

Reviewer comments (10): The section “In Situ Forming Hydrogel Systems” did not show any reference…. why?. The authors can cite: https://doi.org/10.1007/978-981-16-7152-4_3

Response 10: Thank you for the correction and the attached reference. Apologies, the absence of references in the first paragraph was an oversight on our part. The reference above “Sun, X.; Agate, S.; Salem, K. S.; Lucia, L.; Pal, L. Hydrogel-Based Sensor Networks: Compositions, Properties, and Applications - A Review. ACS Appl. Bio Mater. 2021, 4 (1), 140–162. https://doi.org/10.1021/ACSABM.0C01011/ASSET/IMAGES/MEDIUM/MT0C01011_0008.GIF(inline citation [126]) has now been included in lines 472 to 474, with the addition of references [50] and [127] into the first paragraph of Section 3.3 (page 14, lines 471 to 481) as shown below.

Lines 471-481

“In situ forming hydrogels, comprised of three-dimensional polymer networks cross-linked by physical or chemical bonds exhibit remarkable swelling ability in aqueous media whilst attaining structural integrity [126]. Hydrogels exhibit remarkable swelling ability in aqueous media whilst retaining structural integrity [50]. Also, unlike the direct administration of therapeutics via IVI, hydrogel-based drug delivery systems can be designed to control and sustain the localised intraocular release of therapeutics, thus reducing the need for frequent administration. The use of bioerodible hydrogel systems also eliminates the need for subsequent surgical removal procedures, a quality that has become increasingly popular in the field of ophthalmology in recent years, particularly in the treatment of posterior segment pathologies [127].”

Reviewer 3 Report

Comments: I think that the focus of this article is suitable to the overall goal of pharmaceutics/MDPI and this would add value if revised properly.

1-    Although physicochemical properties of HA was mentioned, However, the reaction between HA and ocular inflammation was not written. How inflammatory cytokines (microenvironment) can affect physicochemical properties of HA.

2-    Authors did not observe if HA can be crosslinked strongly during acute or serve inflammation causing agglomeration that could prevent its biodegradation and can obtain cytotoxicity. It suggested strongly if authors can add subsection for describing cytotoxicity of HA.

3-    Although the information gathered in the tables is valuable, it was not properly framed in the text. Indeed, the nanoparticle preparation procedures and the drug encapsulation efficiency have not been described.

4-    Several parameter of HA nanoparticles (g. size, chemical composition and surface charge) that are important to understand their behavior in ocular  systems  were not considered. Moreover, to highlight the putative benefits of targeted therapy using nanoparticles the pharmacokinetics and pharmacodynamics parameters of nanoparticles encapsulated drugs should be presented and compared with those obtained for the corresponding drug under conventional administration.

5-    The manuscript is missing a comprehensive discussion If some formulations have reached clinical application, the authors should state it clearly and discuss the clinical results achieved.

6-    REFs should have been written in the same format style of Pharmaceutics template ((Author 1, A.B.; Author 2, C.D. Title of the article. Abbreviated Journal Name Year, Volume, page range).

7-    In the text, reference numbers should be placed in one square brackets [ ].

Author Response

I would like to take this opportunity to thank you for reviewing our manuscript. The time and effort that you have dedicated to revising our submitted manuscript and providing insightful feedback are greatly appreciated.

1-    Although physicochemical properties of HA was mentioned, However, the reaction between HA and ocular inflammation was not written. How inflammatory cytokines (microenvironment) can affect physicochemical properties of HA.

2-    Authors did not observe if HA can be crosslinked strongly during acute or serve inflammation causing agglomeration that could prevent its biodegradation and can obtain cytotoxicity. It suggested strongly if authors can add subsection for describing cytotoxicity of HA.

Responses 1 and 2: We would like to thank the reviewer for these insightful suggestions. We agree that this section is of particular importance when discussing the biological properties of HA and it was overlooked in the initial manuscript submission. Section 2.5 (page 10, lines 333 to 370) has now been added in which the relationship between HA and ocular inflammation and the potential cytotoxicity of HA under pathological inflammatory conditions are discussed.

Lines 333-370

“2.5  Potential Cytotoxicity of HA under Inflammatory Conditions

The relationship between HA MW and its inflammatory activity has been well documented in literature, particularly in osteoarthritis models. The sequestration of multiple CD44 receptors via NHA and HMWHA-mediated divalent bonding suppresses both the expression of the pro-inflammatory cytokines and the formation of proteoglycans [97].

 Conversely, LMWHA preferentially binds to TLR 2 and 4, transmembrane receptors that are primarily responsible for immune response initiation by pathogen-associated molecular pattern recognition [98]. The complexation of LMWHA and TLR triggers the nuclear translocation of the NF-κB protein, thus triggering the expression of several pro-inflammatory cytokines such as tumour necrosis factor-α (TNF- α), interleukin (IL)-1β, IL-6 and IL-8 [94]. As such, LMWHA fragments serve as potential biomarkers of cellular damage.

Under oxidative stress conditions, an inverse correlation exists between the concentration of HA and the chain length of the respective HA chains at the site of tissue injury [83]. Fragmentation of NHA via reactive oxygen (ROS) and nitrogen (RNS) species production and imbalances in HA synthase and hyaluronidase activity can result in excessive HA degradation via glycosidic bond cleavage, thus forming biologically active, pro-inflammatory HA fragments [99]. A vicious cycle between the mass production of ROS/RNS during tissue injury and pro-inflammatory cytokine activation via LMWHA-receptor binding is established which may further amplify the inflammatory state and disease progression. It is also postulated that the reduction in thickness and viscosity of the HA protective barrier surrounding cells via ROS mediation contributes to both increased cell receptor accessibility and enhancement of the innate immune and inflammatory response [53].

Cytotoxicity can also arise due to the covalent modification of HA resulting from overexpression of tumour necrosis factor-stimulated gene-6 (TSG-6), an inflammation-associated secreted protein induced by TNF- α and IL-1, during inflammation [100]. TSG-6 catalyses the covalent bonding of the heavy chain (HC) domains of the chondroitin sulfate chains of IαI to HA, which results in the sequestration and aggregation of HA chains within a gelatinous HA-HC complex [71, 101, 102]. It is possible that HA-HC complexes can form within ocular tissues, due to the detection of both TSG-6 within human corneal epithelia and IαI within porcine trabecular meshwork cells, respectively [103, 104]. The complexed aggregates are immobile and more viscous in comparison to NHA and as such, exhibit enhanced resistance to biodegradation [105]. However, although such characteristics are favourable for the attenuation of joint degradation in arthritis, the intraocular sequestration of HA-HC aggregates may result in increased intraocular pressure and enhanced inflammation via the enzymatic and chemical fragmentation of localised HA within the complexes.”

3-    Although the information gathered in the tables is valuable, it was not properly framed in the text. Indeed, the nanoparticle preparation procedures and the drug encapsulation efficiency have not been described.

4-    Several parameter of HA nanoparticles (g. size, chemical composition and surface charge) that are important to understand their behavior in ocular systems were not considered. Moreover, to highlight the putative benefits of targeted therapy using nanoparticles the pharmacokinetics and pharmacodynamics parameters of nanoparticles encapsulated drugs should be presented and compared with those obtained for the corresponding drug under conventional administration.

Responses 3 and 4: We agree with the suggestions provided by the reviewer. The physicochemical attributes of HA polyelectrolyte complexes with respect to ocular drug delivery applications were highlighted in Section 4.4 (lines 962 to 1102). However, we agree that such parameters should also be highlighted when discussing HA-based nanoformulations. Therefore, we have modified Section 3.6 (lines 646 to 656 and lines 662 to 710), copied below, to include more information on how some of the physicochemical attributes of HA impacted the pharmacokinetic profiles of the nanoformulations synthesised in the research articles referenced. Some nanomaterial formulation methods have also been briefly discussed and references to reviews focused on the advantages and disadvantages of several formulation methods (References [154] and [155]) have also been included.

Lines 646-656

“As highlighted in Table 2, a variety of formulation methods are used in the synthesis of HA-based nanoformulations. However, preparation methods involving the use of desolvating agents, cross-linking moieties, large volumes of organic solvent or high-energy formulation conditions to formulate nanoparticles with optimal physicochemical attributes that are suitable for mass scale-up are becoming increasingly unfavourable [154, 155]. This is due to the potential cytotoxicity risks imparted by organic residues and the cost of the extensive purification steps to eliminate such residual material. The examples below highlight that HA-based nanoformulations prepared via self-assembly, exhibit considerable efficacy to alternative formulation methods in alleviating adverse conditions in both in vitro and in vivo ocular disease models, with the added advantage of mild formulation conditions.”

Lines 662-710

“3.6.1  Enhanced Pharmacokinetics of HA-based Nanoformulations

The addition of a HA coating can effectively enhance the intraocular migration of nanomaterials, particularly in the vitreous humour due to charge-mediated repulsion between the anionic carboxylate anions of HA and the negatively charged vitreal glycosaminoglycan moieties at physiological pH. HA-coated, mRNA-loaded lipoplexes of less than 100 nm exhibited enhanced mobility through bovine vitreous humour in an ex vivo model in comparison to the positively charged, uncoated lipoplexes [168]. The coating of the lipoplexes with HA with MWs of 22 and 2,700 kDa did not trigger mRNA release in biological media, thus preserving the initial mRNA complexation efficiency of approximately 100%.

Conversely, the adsorption of 120 kDa HA onto the surface of positively charged, human serum albumin nanoparticles resulted in a decrease in fluorescently labelled connexin43 mimetic peptide (Cx43-MP) encapsulation efficiency from 98.4 ± 0.1% to 85.4 ± 3.7%, with a concurrent reduction in zeta potential from 11.4 ± 0.2 mV to -18.2 ± 0.7 mV. The chemical conjugation of HA to nanoparticles, in which CX43-MP was loaded via an incorporation method, also resulted in a reduction in Cx43-MP encapsulation efficiency from 79.0 ± 1.9% to 71.1± 0.8%. However, the chemically modified nanoparticles also exhibited a highly negative surface charge (-44.0 ± 0.4 mV), which ultimately enhanced penetration through both the outer nuclear and RPE layer of an ex vivo bovine retinal model after 4 hours of incubation. Conversely, the intraretinal diffusion of the uncoated nanoparticles was limited to the ganglion cell layer after 2 hours [159]. In comparison to uncoated gold nanoparticles, HA-coated gold nanoparticles exhibited enhanced diffusion from the vitreal injection site within 4 hours post-administration in an ex vivo porcine model [165]. The coated nanoparticles also maintained their inherent red-brown colour within the vitreous humour for up to 24 hours, whereas the uncoated nanoparticles began to aggregate, indicated by a red-brown to purple colour change over the same period [158]. Ex vivo corneal permeability in a murine model increased significantly upon integrating HA within and around liposomal vesicles [169]. Hydrating a thin layer of soybean phosphatidylcholine with 0.7% (w/v) HA resulted in the spontaneous formation of gel-integrated liposomes. As the concentration of HA was fixed at 0.7% (w/v), it was unclear how the addition of HA affected fluconazole (FLZ) encapsulation efficiency. However, FLZ encapsulation efficiency increased from 25.8 ± 4.9% to 56.8 ± 2.5%. with an increase in FLZ loading from 0.3 to 1.2% (w/v). Corneal permeation was enhanced following HA modification, with the HA-modified liposomes allowing for the permeation of over 350 µg/cm3 of FLZ, an amount that was 1.9 and 2.6 times higher than that of the unmodified liposomes (~ 200 µg/cm3) and FLZ suspension (~ 140 µg/cm3), respectively. Stabilisation of the liposome core via HA gel integration also contributed to the sustained release of FLZ in vivo. The creation of a highly viscous, mucoadhesive diffusion network within the liposomes allowed for the sustained delivery of FLZ at a concentration above its minimum inhibitory concentration of 8 µg/mL for over 24 hours post instillation in an in vivo murine model. Conversely, the FLZ suspension reached a Cmax of 60 µg/mL 2 hours post instillation, followed by a rapid reduction in FLZ permeation after 6 hours. Methyl ether poly(ethylene) glycol and 3-amino-1-propanol block copolymer micelles coated with HA exhibited a 1.5-fold increase in genistein corneal penetration in comparison to a genistein eye drop formulation in an ex vivo murine model after 10 hours post-administration, attributed to the increase in precorneal retention time via hydrogen bond-mediated mucoadhesion between HA and mucin chains [170].

3.6.2   Cytotoxicity and Safety of HA-based Nanoformulations”

5-    The manuscript is missing a comprehensive discussion If some formulations have reached clinical application, the authors should state it clearly and discuss the clinical results achieved.

Response 5: We agree with the reviewer that the addition of a section describing the clinical applications of HA would be highly beneficial. We have accordingly added Section 3.7 (page 28, lines 728 to 790) as shown below which details recent clinical applications of HA-based artificial tear and eye drop solutions for the treatment of dry eye disease.

Lines 728-790

“3.7   Clinical Applications of HA-based Ocular Drug Delivery Systems

Due to its remarkable hydrating properties, HA is now the predominant active pharmaceutical ingredient in several commercially available AT formulations (Vismed ®, Blink ® Tears and Hyalein ®) and contact lens multi-purpose solution (BioTrue ®, Hidro Health) [52, 173]. The treatment of moderate to severe dry eye disease (DED) has been the primary focus of clinical trials involving HA-based topical eye drop formulations in recent years. A meta-analysis conducted by Yang et al. [157], in which 19 studies with a total of 2078 cases were evaluated, highlighted the superior efficacy of HA-based artificial tear (AT) formulations in comparison to alternative AT and saline solutions in improving DED symptoms. Similarly, Hynnekleiv et al. [174] have previously summarised the results of 53 clinical trials involving the use of AT solutions containing 0.1-0.4% HA in the treatment of DED over 3 months. The safety and efficacy of HA-based formulations demonstrated in some completed clinical trials with published results will be discussed here.

When compared to autologous serum eye drops derived from the patient’s blood samples, Beck et al. [175] observed that Comfort Shield ®, an eye drop formulation containing 0.15% HMWHA, exhibited comparable efficacy in treating patients with severe, late-stage dry eye disease in an initial, randomised clinical trial. However, it was noted that, although comparable efficacy was achieved between the two formulations, severe corneal surface irregularities limited the reliability of the corneal fluorescein staining score. Also, the authors reported that due to the small size of the cohort (8 in total, 4 – control group, 4 – Comfort Shield ®), further clinical trials with a larger cohort would be required to ensure the validity of the results obtained.

In an investigator-masked, randomised clinical assessment, a combination eye drop formulation containing glycerine and erythritol, carboxymethyl cellulose (0.5%) and HA (0.1%) exhibited comparable efficacy to a 0.18% HA formulation, indicated by a statistically similar reduction in Global Ocular Staining Score from baseline after 3 months [176]. Although both formulations were well tolerated, indicated by minimal changes in best-corrected visual acuity and a low incidence of treatment-related adverse effects, there was a significant decrease in the number of reported cases of subjective DED symptoms (burning sensation, itching etc.) in the combination treatment in comparison to the 0.18% HA formulation. A statistically significant increase in tear film break-up time (TFBT) was reported for patients treated with 0.9% saline solution and 0.1 – 0.3% sodium hyaluronate-based eye drop formulations (Ocudry ®) 30 minutes after instillation in a double-masked, randomised trial [177]. Subjective vision significantly improved post-instillation, with the 0.2% and 0.3% Ocudry ® solutions exhibiting a marked improvement from 1 minute to 20 minutes post-instillation. In terms of comfort, Ocudry ® 0.3% scored significantly less at 1 minute and 5 minutes post-instillation, attributed to the higher viscosity of the formulation with increasing sodium hyaluronate concentration. However, there was a significant improvement in subjective comfort score after 20 minutes from baseline in the Ocudry ® 0.3% cohort.

Pinto-Fraga et al. [178] demonstrated the superior efficacy of Visaid ®, a 0.2% sodium hyaluronate solution in comparison to a 0.9% saline solution in reducing the Ocular Surface Disease Index (OSDI) score after daily instillation of 3-8 drops in a phase II, double-blind clinical trial. Patients treated with 0.2% Visaid ® exhibited a percentage reduction in OSDI index value of -19.5 ± 27.3 % from baseline after 1 month, with over 31.2% of patients reporting an OSDI comfort value of greater than 5 points. No significant changes in intraocular pressure (IOP), TFBT and visual acuity from baseline were reported, thus confirming the tolerability of the formulations. Interestingly, when 0.3% Visaid ® was compared to 0.9% saline in a later phase III double-blind clinical trial, a statistically significant increase in TFBT from baseline was recorded (+ 13.98 ± 26.19 %, p > 0.05) for patients treated with 0.3% Visaid ® [179]. A significant difference in the percentage change of TFBT from baseline was also reported between those treated with 0.3% Visaid ® and 0.9% saline control after 1 month. The MW of the sodium hyaluronate within the Visaid ® solutions was not reported. The MW of the sodium hyaluronate within the Visaid ® solutions was not reported. Assuming the MW of the solutions is similar, it would have been interesting to investigate if a similar therapeutic response could be achieved with a lower dose of the 0.3% Visaid ® solution due to the concentration-dependent increase in viscosity. A recent clinical evaluation was conducted by Alcon Research to evaluate the safety and efficacy of the dual implantation of a CyPass ® Micro-Stent and a commercially available viscoadaptive ophthalmic viscosurgical device (Healon5 ®) in lowering IOP in a cohort with open-angle glaucoma [180]. 83.3% and 73.5% of the patients treated with the CyPass ® Micro-Stent and 30 µL of 60 µL of Healon 5, respectively experienced a reduction in IOP of over 20% from baseline after 12 months post-implantation.”

6-    REFs should have been written in the same format style of Pharmaceutics template ((Author 1, A.B.; Author 2, C.D. Title of the article. Abbreviated Journal Name Year, Volume, page range).

Response 6: Thank you for pointing this out. The references listed in the bibliography have now been updated to correctly reflect the referencing style utilised by Pharmaceutics/MDPI journals.

7-    In the text, reference numbers should be placed in one square brackets [ ].

Response 7: Thank you for this correction. The inline reference numbers have now been updated to include a single square bracket.

Round 2

Reviewer 2 Report

The article can be accepted

Reviewer 3 Report

Manuscript has been revised point by point according to reviewer comment.

Manuscript is more acceptable NOW.